# Improvements in one-dimensional grounding-line parameterizations in an ice-sheet model with lateral variations (PSUICE3D v2.1)

David Pollard[1], Robert M. DeConto[2]

[1]Earth and Environmental Systems Institute, Pennsylvania State University, University Park, PA 16802, USA
[2]Department of Geosciences, University of Massachusetts, Amherst, MA 01003, USA

*Correspondence to*: David Pollard (pollard@essc.psu.edu)

**Abstract.** The use of a boundary-layer parameterization of buttressing and ice flux across grounding lines in a two-dimensional ice-sheet model is improved by allowing general orientations of the grounding line. This and another modification to the model's grounding-line parameterization are assessed in three settings: rectangular fjord-like domains (MISMIP+ and MISMIP3d), and future simulations of West Antarctic ice retreat under RCP8.5-based climates. The new modifications are found to have significant effects on the fjord-like results, which are now within the envelopes of other models in the MISMIP+ and MISMIP3d intercomparisons. In contrast, the modifications have little effect on West Antarctic retreat, presumably because dynamics in the wider major Antarctic basins are adequately represented by the model's previous simpler one-dimensional formulation. As future grounding lines retreat across very deep bedrock topography in the West Antarctic simulations, buttressing is weak and deviatoric stress measures exceed the ice yield stress, implying that structural failure at these grounding lines would occur. We suggest that these grounding-line quantities should be examined in similar projections by other ice models, to better assess the potential for future structural failure.

## 1. Introduction

Accurate modeling of long-term Antarctic Ice Sheet variations requires simulation of ice dynamics in the zone between grounded ice and floating ice shelves, and grounding-line retreat and advance over century and millennial year time scales. Realistic simulation of grounding-line migration is challenging, requiring either higher-order or full-Stokes dynamics (e.g., Seddick et al., 2012), or at least a hybrid combination of horizontally stretching flow (Shallow Shelf Approximation, predominant in shelves and streams) and vertically shearing flow (Shallow Ice Approximation, predominant in inland flow) (e.g., Bueler and Brown, 2009). In any case, sensitivity tests have found that without additional measures, the grounding zone needs to be resolved at fine horizontal resolution on the order of ~100 m to avoid large numerical errors in grounding-line movement (Schoof, 2007; Goldberg et al., 2009; Gladstone et al., 2010, 2012; Pattyn et al, 2012; Cornford et al., 2016). Even with adaptive mesh refinement (Cornford et al., 2013, 2015), long-term $O(10^4$ to $10^6$ year) continental-scale simulations are currently computationally infeasible with this approach. Alternately, the ice flux across grounding lines can be parameterized using an analytic boundary-layer treatment (Schoof, 2007) and embedded in an ice-sheet model (Pollard et al., 2012), making long-term large-scale simulations feasible. This approach performs reasonably well in some idealized model intercomparisons (Docquier et al., 2011; Pattyn et al., 2012; c.f., Gudmundsson, 2013), but less well in others with smaller-scale transient experiments (Pattyn et al., 2013; Pattyn and Durand, 2013; Drouet et al., 2013; Cornford et al., 2020). In this paper we describe new modifications to the parameterized grounding-line flux approach, and show that they significantly improve model performance in some intercomparisons.

Analytic boundary-layer treatments of buttressing and ice velocities across grounding lines (e.g., Schoof, 2007) are usually 1-D, i.e., formulated with one horizontal dimension along the flowline and no lateral variations. In ice-sheet models with two horizontal dimensions, such formulations can be used to prescribe the approximate flow across grounding lines. In our previous work (Pollard and DeConto, 2012; DeConto and Pollard, 2016), this was done simply by applying the 1-D expressions at individual one-grid-cell-wide segments separating pairs of grounded and floating cells, so that the orientation of each single-cell "grounding-line" segment is parallel to either the $x$ or the $y$ axis. Although this is consistent with the one-dimensional character of the formulation in Schoof (2007), it does not capture the actual orientation of the wider-scale grounding line.

Here we implement a more realistic treatment of grounding-line buttressing and ice flow, by applying the 1-D expressions to normal flow across an estimated grounding-line orientation that is not constrained to one or the other grid axes. In principle this is more physically complete than the previous single-cell treatment, and is expected to improve model results. The new grounding-line orientation determines the direction of the ice flux; however, in the calculation of buttressing, overall best results are obtained by using the minimum buttressing over all possible directions, as described below.

Three types of experiments are used to assess the above modifications. First, simulations are performed for a fjord-like glacier confined to a relatively narrow channel, as in the MISMIP+ intercomparison (Cornford et al. 2020). Because of the confining lateral boundaries and a central bedrock depression, grounding lines in these simulations have large two-dimensional curvatures, and provide a good test for the changes implemented here. Second, results are shown for the MISMIP3d intercomparison (Pattyn et al., 2013), also in a fjord-like setting. Third, much larger-scale simulations of future ice retreat in West Antarctica are performed, forced by warming climates corresponding to the extreme RCP8.5 greenhouse gas emissions scenario.

In section 2, the modifications to the buttressing and grounding-line flux parameterizations are described in detail. Sections 3 and 4 present results for the fjord-like MISMIP+ and MISMIP3d experiments, respectively. Section 5 presents results of the West Antarctic future simulations. In section 6, deviatoric stresses at grounding lines in West Antarctic simulations (without hydrofracturing or cliff-failure physics) are examined, as they retreat across very deep bedrock topography in central West Antarctica in future centuries, to assess the potential for structural failure that could lead to very rapid disintegration of the remaining ice. In Appendix A, four alternate calculations of buttressing are compared to the omni-directional treatment used in the main paper, showing that the latter yields best overall results for MISMIP+ and MISMIP3d, but they all have very minor effects in the West Antarctic simulations. In Appendix B, three additional and more speculative modifications to the model's grounding-line flux parameterization are described. Finally, Appendix C describes a minor change used here in the calculation of crevasse depths, based on principal deviatoric stress rather than divergence, which is a small improvement "in principle" but is shown to have insignificant effects in the Antarctic simulations.

## 2. Methods

As described in Pollard and DeConto (2012), the primary grid in the finite-difference ice-sheet model is the $h$-grid, with ice thicknesses ($h$) defined at the center of each cell. Ice in each $h$-grid cell is either floating in the ocean or is grounded, depending on the ice thickness, bedrock elevation, and sea level. At the grid-cell level, the boundary between floating and grounded-ice regions consists of piecewise-linear segments at the edges between pairs of $h$-grid cells, with each edge parallel to the $x$ or $y$ axis

(Fig. 1a). The model uses an Arakawa-C grid, in which horizontal $u$ and $v$ velocities are staggered half a grid cell in the $x$ and $y$ directions respectively, so each segment separating floating or grounded $h$-cells has a $u$ or $v$ velocity defined at its mid-point, as indicated in Fig. 1a. (The model performs a sub-grid interpolation that refines the grounding-line position between each pair of $h$-grid cells and does not coincide with the cell edge between them, but that does not affect the material presented here).

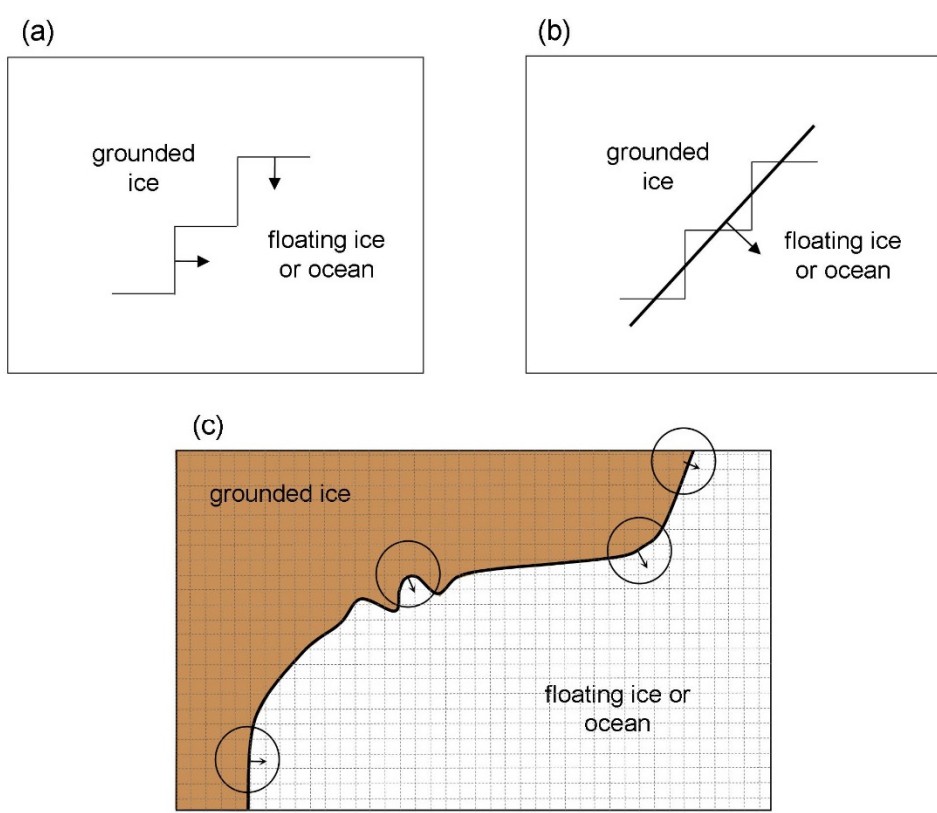

**Figure 1.** Schematics of grounding-line orientation treatment. Edges of $h$-grid cells are shown by thin lines, with grounded ice on one side (upper left) and floating ice or open ocean on the other (lower right). Ice velocities across grounding lines are shown by arrows. **(a)** Old single-cell piecewise scheme used in previous model versions. **(b)** New scheme with more realistic grounding-line orientation (thicker line). **(c)** New scheme at a larger scale, with the normal at each point determined by the direction towards the center-of-mass of floating ice/ocean points within a given radius. Typical model grid cells are shown by dashed lines.

The model ice dynamics uses a hybrid combination of vertically integrated shallow ice and shallow shelf approximations (SIA, SSA), with the seaward ice flux at grounding lines imposed as a boundary condition according to an analytical expression relating ice flux to ice thickness (Schoof, 2007):

$$q_g = \left(\frac{A(\rho_i g)^{n+1}(1-\rho_i/\rho_w)^n}{4^n C}\right)^{\frac{1}{m+1}} \theta^{\frac{n}{m+1}} h^{\frac{m+n+3}{m+1}} \tag{1a}$$

$U_g = q_g / h$ (1b)

where $q_g$ is the ice flux and $U_g$ is the ice velocity across the grounding line, and $h$ is ice thickness at the grounding line. $\rho_i$ and $\rho_w$ are the densities of ice and ocean water, respectively, and $g$ is gravitational acceleration. $A$ is the rheological coefficient and $n$ is the exponent for ice deformation. $C$ is the coefficient and $m$ is the exponent for basal sliding (Schoof, 2007), written as $C_s$ and $m_s$ in Pollard and DeConto (2012). The term $\theta$ in (1a) represents buttressing by ice shelves, i.e., the amount of back stress caused by pinning points or lateral forces on the ice shelf further downstream. The buttressing factor $\theta$ is defined as the ratio of vertically averaged horizontal deviatoric stress normal to the grounding line, relative to its value if the ice shelf was freely floating with no lateral constraints and no back stress. (The latter free-floating value is always extensional, balancing the difference between the column-mean hydrostatic ice pressure at the grounding line with the smaller mean horizontal component of ocean-water pressure on the ice shelf. Pinning points or lateral forces on the ice shelf reduce this value towards zero, i.e., less extensional and more compressive, so $\theta = 1$ for unbuttressed grounding lines and diminishes towards 0 as buttressing increases.)

The analysis for grounding-line flux and buttressing in Schoof (2007) is limited to one-dimensional flowline geometry. In our previous "standard" model (Pollard and DeConto, 2012), Eq. (1) is applied across individual one-grid-cell-wide segments separating pairs of grounded and floating grid cells, so that the orientation of each single-cell "grounding-line" segment is parallel to either the $x$ or the $y$ axis, as sketched in Fig. 1a. In the standard model, the buttressing factors $\theta_u$ and $\theta_v$ in the $x$ and $y$ directions respectively are:

$$\theta_u = \frac{4\,\eta(\partial u/\partial x)\,h}{\rho_i(1-\rho_i/\rho_w)gh^2/2} \tag{2a}$$

$$\theta_v = \frac{4\,\eta(\partial v/\partial y)\,h}{\rho_i(1-\rho_i/\rho_w)gh^2/2} \tag{2b}$$

where $\eta$ is the non-linear strain-dependent ice viscosity, and the numerators in (2a,b) are 2x the deviatoric stress (times ice thickness $h$) in the $x$ or $y$ directions.

Although this previous treatment of $\theta$ is consistent with the one-dimensional character of the formulation in Schoof (2007), it does not capture the wider-scale orientation of the real grounding line, which does not actually run along the "staircase" single-cell segments as in Fig. 1a. A new method for the direction of $U_g$ and the value of $\theta$ is described below, and sketched in Fig. 1b and 1c. It allows for general grounding-line orientations running at an angle to the grid axes, and applies the ice flux given by (1) in a direction normal to this grounding line. First, an estimate of the grounding-line orientation is needed, that represents a spatial smoothing of the boundaries of nearby cells. A simple algorithm is used, as follows.

(i)  Consider all grid cells within a given radius $R_c$ of the location in question $(x_c, y_c)$, and take the average of the $x$ and $y$ coordinates of cells with ocean or floating ice (not grounded ice), $(x_o, y_o)$. If this radius extends beyond the domain boundaries, virtual points are used with their grounded or floating property equal to that extended normally from the domain boundary.

(ii) Then the normal to the grounding line (in the direction towards the ocean) is $(x_o - x_c, y_o - y_c)$. The length of this vector is normalized to 1 meter, and is called $(n_x, n_y)$ below.

The resulting grounding-line orientations in some MISMIP+ experiments are shown below, which show that the algorithm works as expected. The choice of radius $R_c$ distinguishes small-scale sinuosities in the grounding line that are averaged out, and larger-scale curvilinear features that should be retained. For the relatively confined fjord MISMIP+ experiments below, $R_c$ is set to 20 km, and for the much larger-scale Antarctic simulations it is set to 50 km. In sensitivity tests (not shown), choices of $R_c$ between 10 to 50 km make very little difference to the results in both types of experiments.

This orientation is used for the direction of the grounding-line velocity (Eq. 7 below). It can also be used in the calculation of $N$, the net deviatoric stress normal to the grounding line, and hence $\theta$. The equations below follow Gudmundsson (2013, his Eqs. 2, 6 and 12).

$$N = \hat{\boldsymbol{n}}^T . (\boldsymbol{T} . \hat{\boldsymbol{n}}) \tag{3}$$

where $\boldsymbol{T}$ is the deviatoric stress tensor (Gudmundsson, 2013) and $\hat{\boldsymbol{n}}$ is the unit vector $(n_x, n_y)$ normal to the grounding line provided by the algorithm above. Expanding in $x,y$ coordinates, this is:

$$N = (2\tau_{xx} + \tau_{yy}){n_x}^2 + 2\tau_{xy}n_x\,n_y + (2\tau_{yy} + \tau_{xx}){n_y}^2 \tag{4}$$

where $\tau_{ij}$ are the 2D components of the stress tensor, obtained from the corresponding strain rates and viscosity $\eta$ (e.g., Thoma et al., 2014):

$$\tau_{xx} = 2\eta\, \partial u/\partial x, \qquad \tau_{yy} = 2\eta\, \partial v/\partial y, \qquad \tau_{xy} = 2\eta\, (\partial u/\partial y + \partial v/\partial x)/2 \tag{5}$$

These velocities $u,v$ are obtained from a preliminary solution of the SSA dynamical equations performed at each timestep without any Schoof-imposed constraints at the grounding line (Pollard and DeConto, 2012), called the "grid-solution" below. Then the buttressing factor $\theta$ is given by

$$\theta = \frac{N}{\rho_i(1-\rho_i/\rho_w)gh/2} \tag{6a}$$

The denominator is the net normal deviatoric stress that would result for a freely floating and completely unbuttressed ice shelf (or a vertical ice face with no ice shelf at all).

In Appendix A, results are shown for several variations in calculating $N$ in (4) and $\theta$ in (6a). These alternatives stem from the inherent uncertainty in using a 1-D flowline parameterization (Eq. 1) within a 2-D model, and we use the MISMIP+ and MISMIP3d results as an empirical guide. The best overall intercomparison results are obtained not with the above method using the single direction $(n_x,n_y)$ in (4), but using the maximum extensional (principal) stress $N_{max}$, i.e., the maximum of $N$ over all possible directions 0 to 360°, and then

$$\theta = \frac{N_{max}}{\rho_i(1-\rho_i/\rho_w)gh/2} \tag{6b}$$

For all new-model results in the main paper, $N_{max}$ is used and $\theta$ is given by (6b). A rationale for this method is discussed in Appendix A, but we emphasize that the choice is guided mainly because it yields the best overall MISMIP+ and MISMIP3d results among all variations tried (Fig. A1). Note also that $N_{max}$ is used only at the grounding line. In the ice-shelf interior, $\theta$ has no effect on the model physics, and where it is shown diagnostically below, the ice velocity at each point provides the orientation in (4).

The value of $\theta$ from (6a) or (6b) can be less than 0 or greater than 1 (Gudmundsson, 2013) as shown in the figures below. However, when used in Eq. (1) to obtain the imposed flow across the grounding line $U_g$, it is restricted to the range [0,1], i.e, reset to max (0, min (1,$\theta$)). Finally, $U_g$ is resolved into its $x$- and $y$-axis components, using the orientation $(n_x, n_y)$ from the algorithm above:

$$u_g = U_g\, n_x \tag{7a}$$

$$v_g = U_g\, n_y \tag{7b}$$

These velocity components are imposed in the final SSA solution at each time step, at staggered $u$ or $v$-grid points as appropriate located at the mid points between pairs of grounded and floating $h$-grid cells. (It is easy to show that this decomposition of $U_g$ onto the $u$ and $v$-grids results in the physically correct net flux across the actual grounding line, averaged over many $u$ and $v$-grid points).

As well as entering in the Schoof grounding-line flux Eq. (1a), the buttressing factor $\theta$ also influences the effects of crevasses and hydrofracturing in grounding-zone cliff-failure (Pollard et al., 2015). These physics are not enabled for all MISMIP+ runs and most of the Antarctic runs below.

## 3. Results: MISMIP+ experiments

As a first test of the modifications above, we use the MISMIP+ experiments (Cornford et al., 2020). These simulate glacier flow in a rectangular fjord-like channel, and involve significant two-dimensional curvatures of grounding lines. The channel is 80 km wide, with bedrock generally sloping downstream and an ice shelf flowing into the ocean. There is a bedrock depression at mid-fjord around $x \approx 400$ km, and a ridge at $x \approx 505$ km, as shown in Fig. 2. All prescribed fields and model solutions are laterally symmetric about the centerline of the channel. Starting from a close-to-equilibrated control state with the centerline grounding line just downstream of the bedrock depression, prescribed perturbations to sub-ice oceanic melt rates (which are zero in the control) are applied for 100 years, and either maintained or re-set to zero for the next 100 years. In the MISMIP+ Ice1 experiment, the applied oceanic melt rate is a smooth function of ice-shelf draft and ocean depth, and in the Ice2 experiment, it is a large uniform value in the downstream section of the fjord (Cornford et al., 2020). The resulting variations of the grounding line are examined, mainly its position along the centerline of the channel. All MISMIP+ runs here use a model resolution of 1 km; results at 2 km are very similar. At 5 km and coarser resolutions, in some runs the curvilinear features in the fjord are not adequately resolved (with only 8 grid points or less in each channel half-width), and results are physically unreasonable.

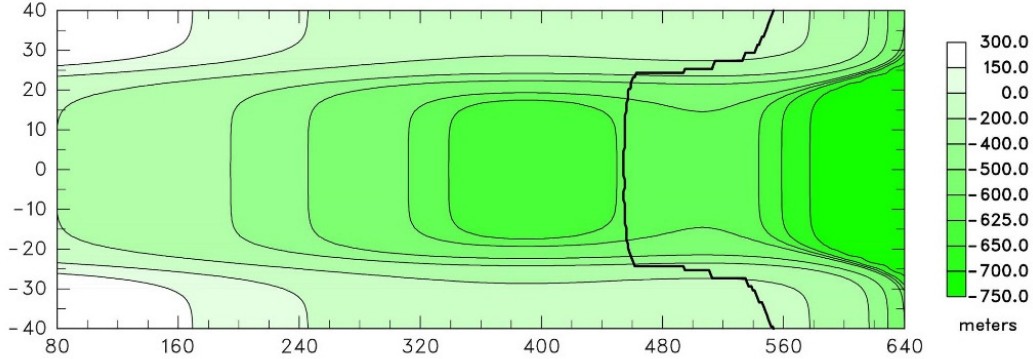

**Figure 2.** Bedrock topography used in the MISMIP+ experiments (Cornford et al, 2020). Also shown is the "control" grounding line after spin-up at year 0 (thick black line). Axes scales are in km. The first 80 km of the channel is not shown. Note the nearly 3x stretching of the channel width relative to the length.

Fig. 3 shows results for the MISMIP+ Ice1 experiment, comparing the new model version with our previous standard model; the
180 latter is very close to that used in the original intercomparison (Cornford et al., 2020). Different values of rheologic coefficient *A* are used as noted in the caption, in order for the equilibrated grounding line at the start of each experiment to have nearly the same *x*-axis location (~455 km). With the previous model version and original MISMIP+ *A* value (thin black lines), the grounding-line variations are close to those in our original MISMIP+ runs, significantly faster and larger than other higher-order, higher-resolution models as shown in Cornford et al. (2020). With the same model version and a reduced value of *A* (crosses),
the results are within the other-model envelopes (background shading), but close to their outer edges; this dependence on *A* in our model was not noticed before. With the new model version and an intermediate value of *A* (thick lines), the grounding-line variations are considerably less rapid and have smaller amplitudes, and lie well within the envelopes of the other higher-order, higher-resolution models in the intercomparison. This suggests that the modifications above are real physical improvements to our model.

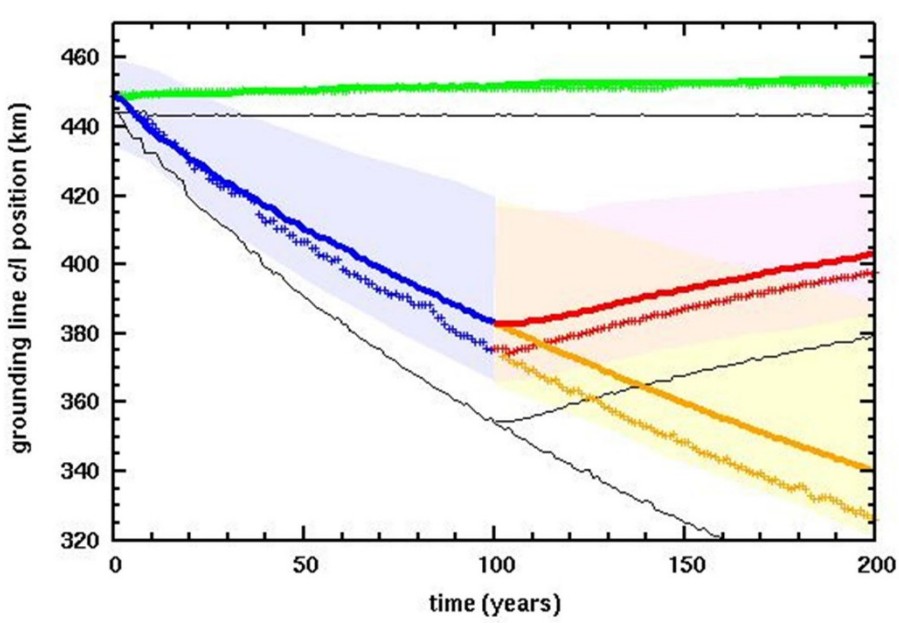

**Figure 3.** Along-fjord centerline position along the *x*-axis (km) of grounding lines in the MISMIP+ Ice1 experiments (Cornford et al., 2020). **Thick colored lines:** new model version and rheologic coefficient $A = 3 \times 10^{-17}$ Pa$^{-3}$ a$^{-1}$. **Crosses:** previous model version and $A = 2.5 \times 10^{-17}$ Pa$^{-3}$ a$^{-1}$. **Thin black lines:** previous model version and $A = 3.5 \times 10^{-17}$ Pa$^{-3}$ a$^{-1}$. **Green:** control, with zero oceanic melt. **Blue and yellow:** with oceanic melt perturbation. **Red:** with oceanic melt reset to zero after year 100. Shaded regions show the envelopes for the "main subset" of MISMIP+ models, copied from Cornford et al. (2020, their Fig. 7a).

Spatial maps of ice extents, grounding lines, and buttressing factors are shown in Fig. 4 for the new model version, at the beginning and end of each 100-year segment. Away from the margins, the grounding-line configurations are quite similar to those for other models shown in Cornford et al. (2020); however, near the margins our grounding lines extend further downstream (to ~550 km) than most other models (~490 to 520 km). The buttressing factors in the right-hand column, not adjacent to the grounding line, are purely diagnostic and have no effect on the model physics. As noted above, they are computed from Eqs. (4) to (6a) using the direction of ice flow as the normal vector in Eq. (4) (cf. Fürst et al., 2016).

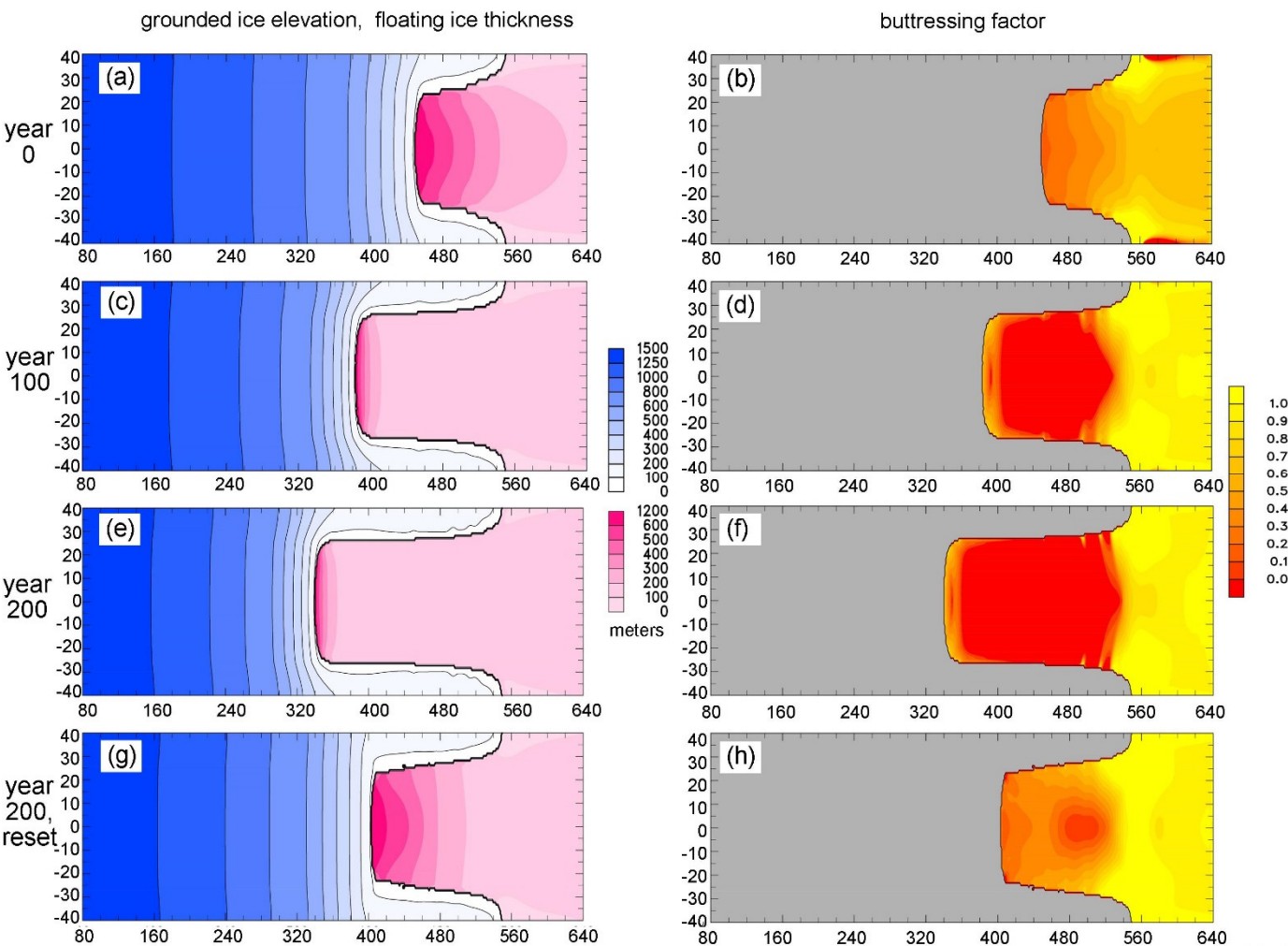

**Figure 4.** Spatial maps in the MISMIP+ Ice1 experiments, for the new model version (as in Fig. 3 with $A = 3 \times 10^{-17}$ Pa$^{-3}$ a$^{-1}$). Flow is left to right. The grounding line is shown by a thick black line. The axes scales (km), truncation of first 80 km, and stretched width are as in Fig. 2. **1st row (a-b):** at year 0 (control). **2nd row (c-d):** at year 100 with oceanic melt perturbation. **3rd row (e-f):** at year 200 with oceanic melt perturbation. **4th row (g-h):** at year 200 with oceanic melt reset to zero after year 100. **1st column (a,c,e,g):** Grounded ice surface elevations (m, blue scale), and floating ice thicknesses (m, pink scale). **2nd column (b,d,f,h):** Buttressing factor $\theta$ (diagnostic except at grounding line).

Fig. 5 shows buttressing factors and grounding-line orientations, as in Fig. 4 but just at the grounding lines. The right-hand panels (buttressing factors) compare favorably with similar plots in Gudmundsson (2013, his Fig. 2). The left-hand panels (grounding-line orientations) shows that the simple algorithm described in section 2 works well and yields appropriate angles for these geometries.

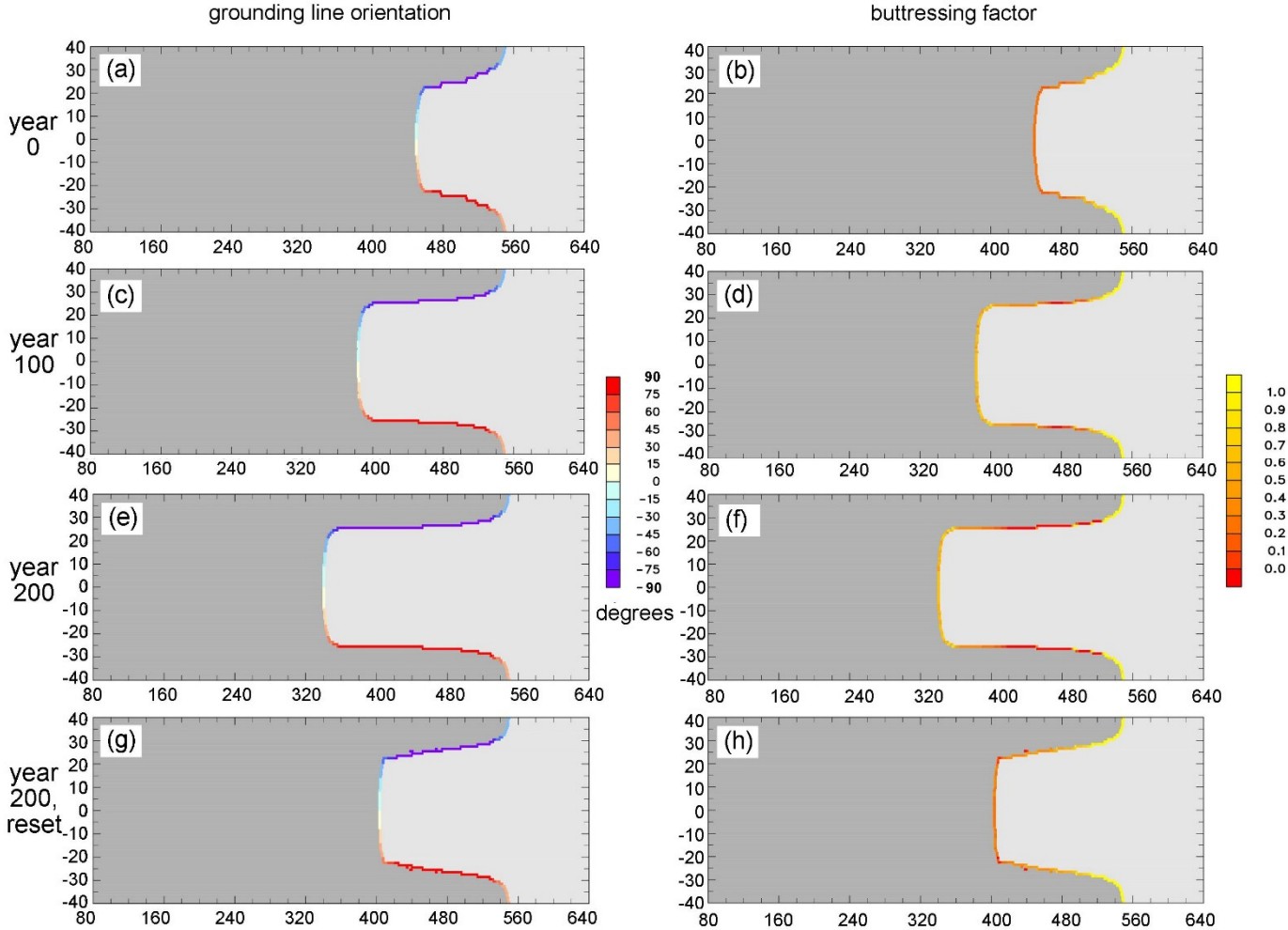

**Figure 5.** As Fig. 4 for quantities at grounding lines. **1st column (a,c,e,g):** Orientation of grounding line (degrees counterclockwise of normal vector $(n_x, n_y)$ from the along-fjord $x$ axis, given by algorithm in section 2). **2nd column (b,d,f,h):** Buttressing factor $\theta$.

Centerline grounding-line variations for the Ice2 MISMIP+ experiment, with a spatially abrupt oceanic melt pattern, are shown in Fig. 6 for the same model versions as in Fig. 3. The modifications have similar effects as in Fig. 3 for the Ice1 experiment, reducing the rapidity and amplitude of the grounding-line variations compared to the previous model version with the original rheologic coefficient $A$ (thin black lines). However, results with the previous model version and reduced $A$ (crosses) and the new model version with intermediate $A$ (thick colored lines) are nearly the same here, and both lie well within the envelopes of other MISMIP+ models (Cornford et al., 2020).

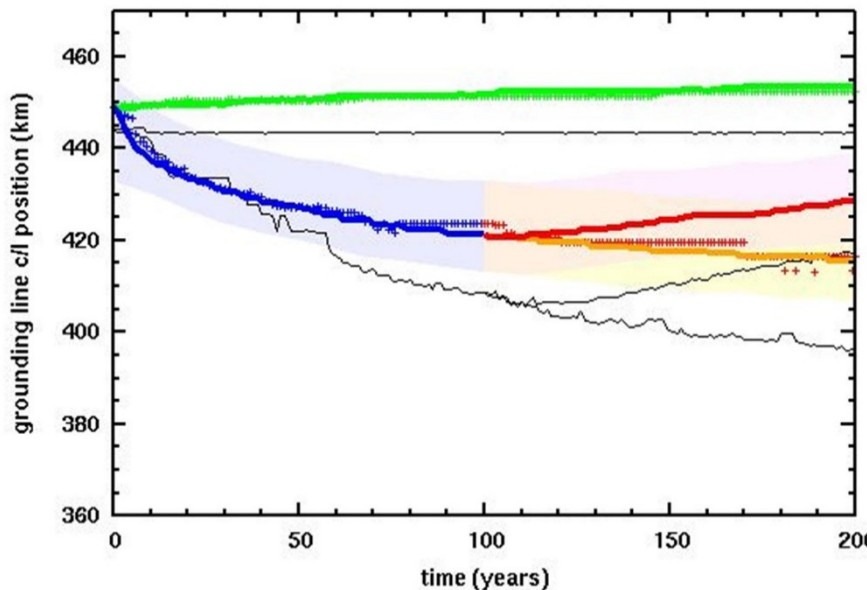

**Figure 6.** As Fig. 3 except for the MISMIP+ Ice2 experiments. Shading for the "main subset" of MISMIP+ models is copied from Cornford et al. (2020, their Fig. 13b).

## 4. Results: MISMIP3d experiments

The MISMIP3d intercomparison (Pattyn et al., 2013) offers another useful test of the new model versions. It uses a rectangular fjord-like setting as in MISMIP+, but with a uniformly sloping bed and perturbations in basal sliding coefficient instead of ocean melting. The models are first run to equilibrium, then the basal sliding coefficient is increased (slipperier bed) in a central region for 100 years causing the grounding line to advance, after which the perturbation is removed. Similarly to MISMIP+, our previous model produced larger and more rapid grounding-line advances than most other higher-order and/or higher-resolution

models in the intercomparison (Pattyn et al., 2013), and consequently the changes in total volume over flotation and cavity volume differed from most models (Pattyn and Durand, 2013).

Fig. 7a,b shows the main results for the MISMIP3d experiment, for the new model version (solid lines) and the previous standard version close to that used in the original intercomparison. The centerline grounding line excursions in Fig. 7a for the new model version are considerably less than previously (~20 km vs. ~30 km), and much closer to the range of other model categories (red

bar on the y-axis, from Pattyn and Durand, 2013). Notably, the equilibrated starting position of the grounding line is now around 560 km, much closer to those of most higher-order models in the intercomparison (~540 km, Pattyn et al., 2013; Pattyn and Durand, 2013). Changes in total volume over flotation and cavity volume in Fig. 7b are also much closer to the ranges of the other model categories (yellow and blue bars on the y-axis; Pattyn and Durand, 2013).

For completeness, spatial maps of changes in surface speed and elevation are shown in Figs. 7c-f, which can be compared with

the same quantities for other model categories in Pattyn and Durand (2013) Figs. 2 and 3. There are some differences but the overall features and amplitudes are similar.

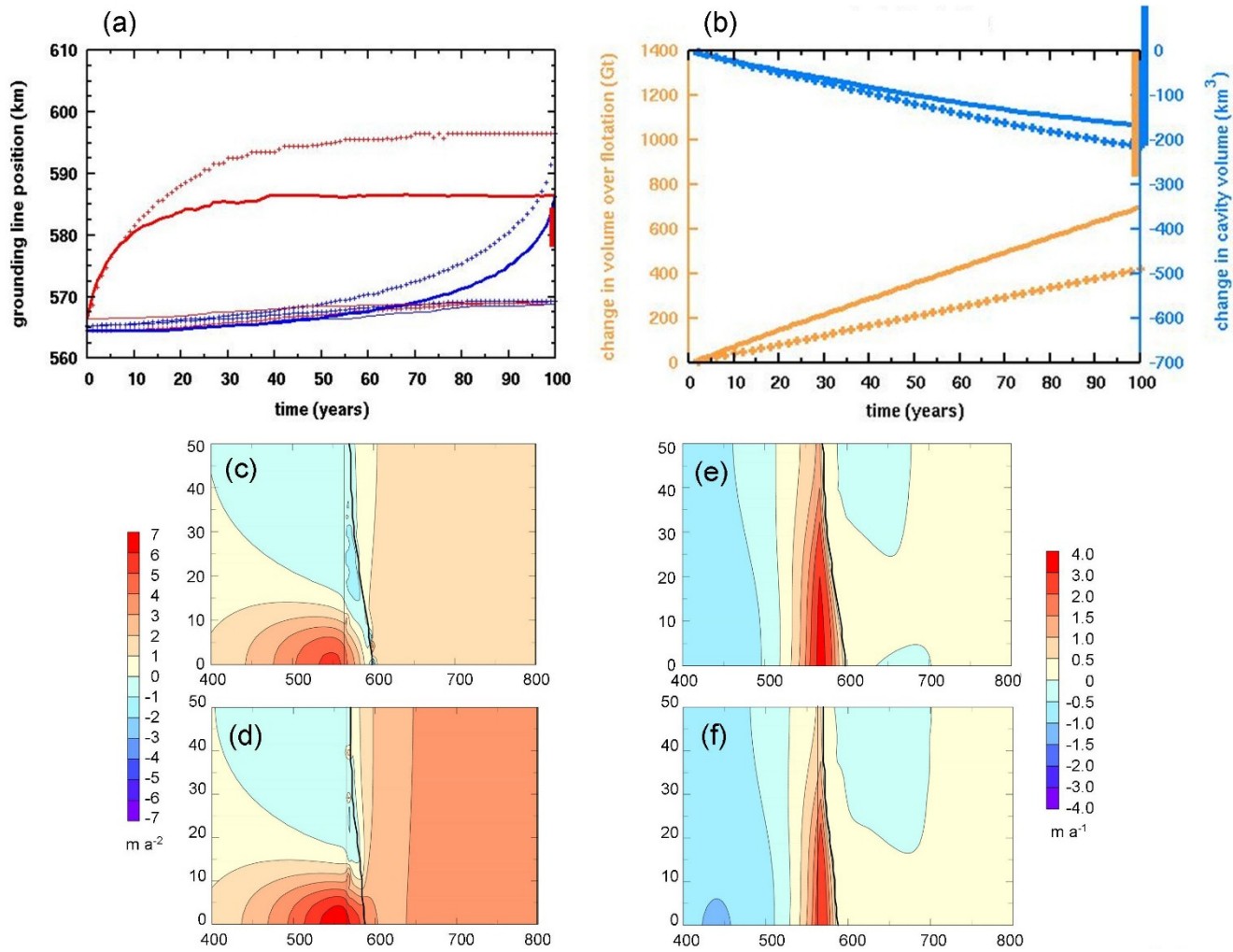

**Figure 7. (a)** Along-fjord position along the domain x-axis (km) of grounding lines in the MISMIP3d experiment (Pattyn et al., 2013). **Solid lines:** new model version. **Crosses:** previous model version. **Red:** advancing grounding lines for 100 years after the perturbation. **Blue:** retreating grounding lines after the perturbation is removed (reverse time axis). Thick upper lines show centerline positions, and thin lower lines show positions at the domain edge. The vertical red bar on the right-hand y-axis shows the range of amplitude of centerline excursions at 100 years for other model categories in the intercomparison (from Pattyn and Durand, 2013, Fig. 1). **(b) Yellow:** Changes in total ice volume over flotation (multiplied by ice density, Gt of ice), for a half-domain from the centerline to one *y*-axis edge (as in Pattyn and Durand, 2013). **Blue:** Changes in total cavity volume under floating ice (km$^3$), for the same half-domain (ibid). Solid lines vs. crosses denote new vs. previous model versions as in (a). The vertical yellow bar on the right-hand y-axis shows the volume-over-flotation range at year 100 for the other model categories in the intercomparison, and the vertical blue bar shows the same range for cavity volume (from Pattyn and Durand, 2013, Fig. 4). **(c,d):** Mean rate of change in ice surface speed from year 0 to year 100 (m a$^{-2}$) as in Pattyn and Durand (2013, Fig. 2). **(e,f):** Mean rate of change in ice surface elevation from year 0 to year 100 (m a$^{-1}$) as in Pattyn and Durand (2013, Fig. 3). **(c,e)** are for the previous model version, and **(d,f)** are for the new model version. In (c-f), thin and thick black lines show the position of the grounding line at years 0 and 100, respectively.

## 5. Results: West Antarctic simulations

To test the modifications in real-world scenarios at larger scales than the idealized fjord experiments above, we simulate retreat of the West Antarctic ice sheet due to future climate warming. The climate forcing follows that in DeConto and Pollard (2016)

for the extreme RCP8.5 greenhouse gas emissions scenario, with atmospheric temperatures and precipitation from regional climate model simulations, and oceanic temperatures from a transient future simulation with the NCAR CCSM4 global climate model (Shields et al., 2017). The ice sheet is initialized to modern observed (Fretwell et al., 2013), and run from 1950 CE for 500 years. A nested domain is used spanning West Antarctica with a polar stereographic grid of 10 km resolution, and with lateral boundary conditions supplied by an earlier continental-scale run.

The mechanisms of hydrofracturing and cliff-failure (Pollard et al., 2015; DeConto and Pollard, 2016) are disabled in the main simulations below, so the future collapse of West Antarctica is relatively slow and driven mainly by sub-ice-shelf oceanic melt and ductile processes as in other models (Feldmann and Levermann, 2015; Golledge et al., 2015; Arthern and Williams, 2017). This provides a better test of the modifications above, without the overall retreat being dominated by more drastic retreat mechanisms.

Fig. 8 shows the equivalent sea level rise corresponding to net ice melt from West Antarctica, for three types of simulations: (1) control with perpetual modern climate, (2) future RCP8.5 scenario with hydrofracturing and cliff collapse disabled, and (3) future RCP8.5 scenario with those mechanisms enabled. Each simulation is run for the same pair of model versions as for the MISMIP+ and MISMIP3d experiments above: the previous standard version and the new version with the modifications described in section 2.

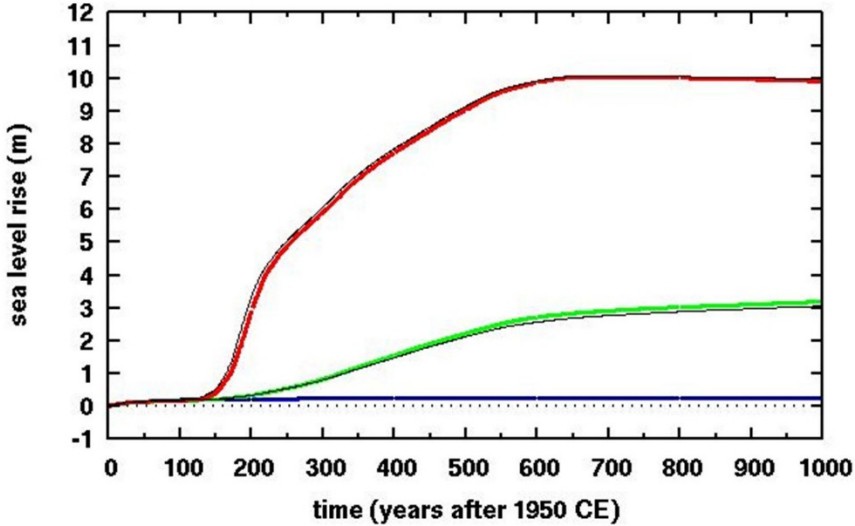


**Figure 8.** Equivalent global sea level rise in simulations of future West Antarctic ice retreat with climate forcing based on the RCP8.5 greenhouse gas scenario. The sea-level rise calculation accounts for ice grounded below sea level, which if melted contributes only its ice-over-flotation amount. **Thick colored lines:** new model version. **Thin black lines:** previous model version. **Blue:** control (perpetual modern climate). **Green:** with RCP8.5 forcing, without hydrofracturing or cliff failure. **Red:** with RCP8.5 forcing, with hydrofracturing and cliff
failure.

As expected, for the future RCP8.5 simulations with no hydrofracturing or cliff failure (type 2), West Antarctic grounding lines retreat deep into the interior over several centuries. After 500 years, nearly all West Antarctic marine ice melts producing ~3 m of sea level rise, similar to that found by the other models noted above. With hydrofracturing and cliff failure enabled (type 3),

much more rapid and pervasive grounding-line retreat occurs, with most West Antarctic marine ice melted within ~200 years, as in DeConto and Pollard (2016). In all simulations, the new modifications make very little difference to these results, in contrast to the MISMIP+ fjord-like experiments. Presumably this is due to the larger lateral scales and less influence of lateral boundaries in the major West Antarctic basins, so that the flow in the central regions of these basins is more 1-D (flowline) in character, better represented by the simpler "staircase" grounding-line treatment of the standard model. This is consistent with our results in the ABUMIP intercomparison involving continental Antarctic experiments, where the previous model version was used and results lie within the ranges of the other models (Sun et al, 2020).

Spatial maps of ice distribution and buttressing factor are shown in Figs. 9 and 10 for selected times in the future simulation without hydrofracturing or cliff failure (type 2). Except for points right at the grounding line, the buttressing factor $\theta$ in Fig. 10 is purely diagnostic and has no effect on the model physics. Away from the grounding line, $\theta$ is calculated for these figures based on stress normal to the direction of ice flow, otherwise following Eqs. (3) to (6b) in section 2 above. There are some differences due to the new modifications, mainly in the ice-shelf interiors for $\theta$ away from grounding lines), but overall the distributions are similar. For modern, the $\theta$ maps can be compared directly with those in Fürst et al. (2016), who calculated the same quantity (their $K_n$ is our $1-\theta$) from assimilated modern ice-shelf velocities, but using the orientation with minimum $N$ (maximum buttressing) instead of ice flow direction. Even so, the patterns compare favorably with our map for the new model version (Fig. 10b).

previous version     new version

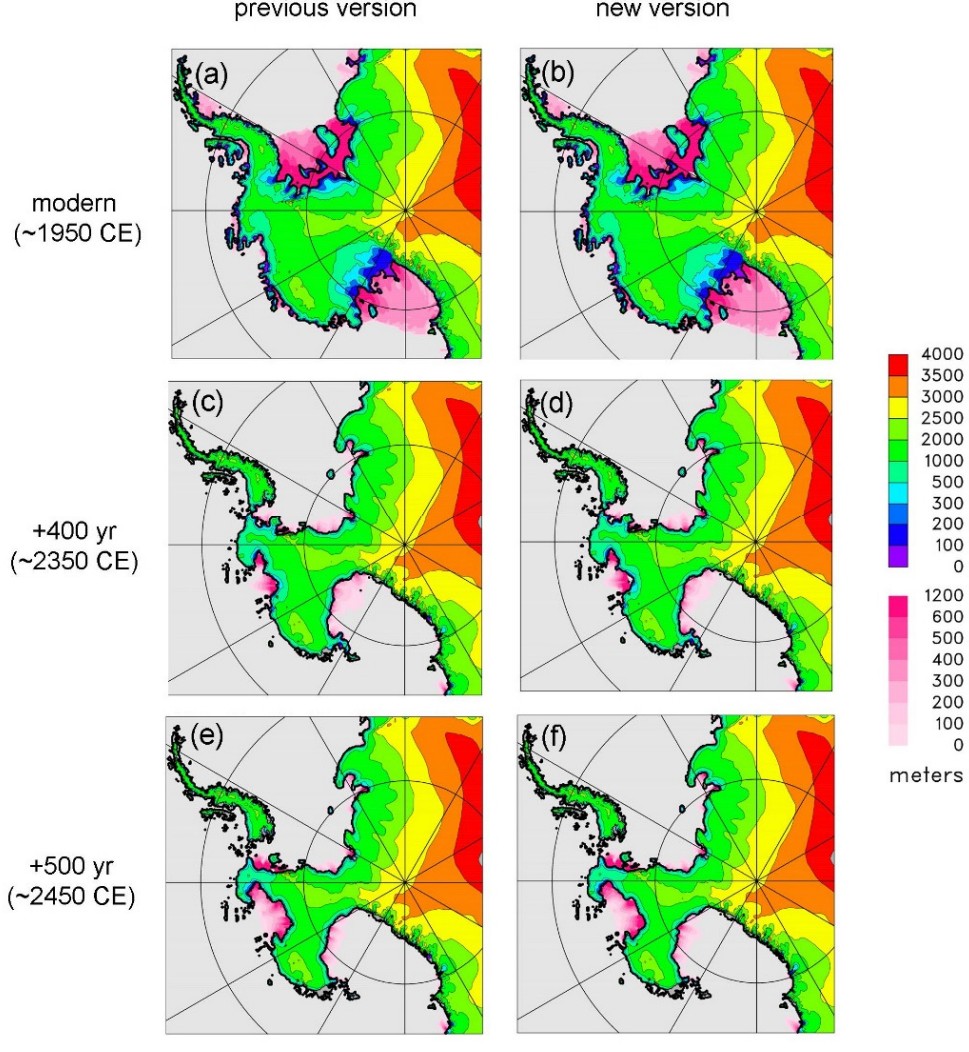

modern
(~1950 CE)

+400 yr
(~2350 CE)

+500 yr
(~2450 CE)


**Figure 9.** Spatial maps of simulated future West Antarctic ice retreat with RCP8.5 forcing, without hydrofracturing or cliff failure, showing grounded ice surface elevations (m, rainbow scale) and floating ice thicknesses (m, pink scale). **1st row (a-b):** at year 0 (~1950 CE). **2nd row (c-d):** at year 400 (~2350 CE). **3rd row (e-f):** at year 500 (~2450 CE). **1st column (a,c,e):** previous model version. **2nd column (b,d,f):** new model version.

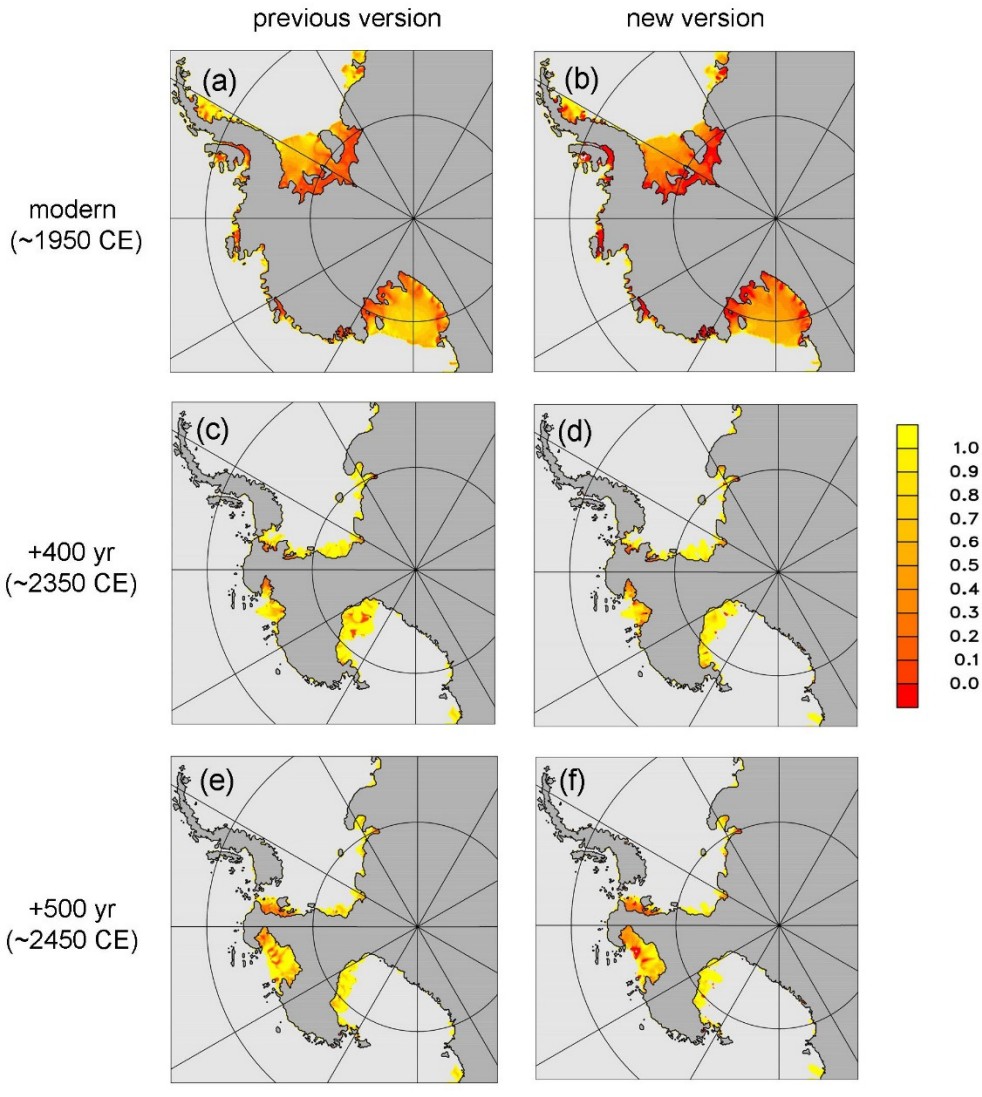

previous version | new version

(a) (b) modern (~1950 CE)

(c) (d) +400 yr (~2350 CE)

(e) (f) +500 yr (~2450 CE)

1.0
0.9
0.8
0.7
0.6
0.5
0.4
0.3
0.2
0.1
0.0


**Figure 10.** As Fig. 9 showing buttressing factor $\theta$ (diagnostic except at grounding line).

## 6. Results: potential for structural failure at West Antarctic grounding lines

As grounding lines retreat across central West Antarctica in the RCP8.5-driven simulations above, they encounter very deep bathymetry with depths of ~1 to 2.5 km below sea level, especially in the Bentley Subglacial Trench (Fig. 11). Simple vertically

integrated force balance calculations (Bassis and Walker, 2012; Pollard et al., 2015) and vertically resolved modeling (Bassis and Jacobs, 2013; Ma et al., 2017; Schlemm and Levermann, 2019; Benn et al., 2019; Parizek et al, 2019; cf. Clerc et al., 2019) suggest that ice columns at such deep grounding lines, if unbuttressed or only weakly buttressed by ice shelves or mélange, will be structurally unstable, with deviatoric stresses exceeding the material yield stress of the ice. Once initiated, structural "cliff" failure would be expected to propagate extremely rapidly into ice upstream of the grounding line, only stopping when shallower

bathymetry is reached, or if buttressing increases somehow.

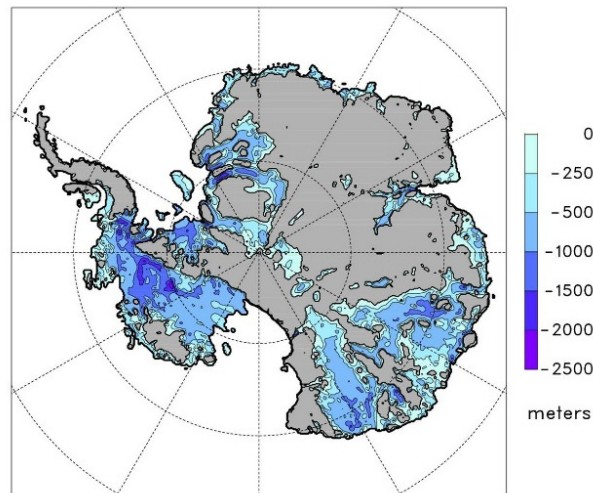

**Figure 11.** Modern observed Antarctic bedrock elevations where below sea level, aggregated to the 10-km model grid from Bedmap2 (Fretwell et al., 2013). As in Fig. 1b of Pollard et al. (2015).

In our simulations without hydrofracturing or cliff failure physics (type 2), structural failure is not part of the model, but we can use the new improved calculations of grounding-line buttressing factors and deviatoric stresses to examine the basic force balance as grounding lines traverse the deep central West Antarctic regions, and so to diagnose if structural failure would occur, or conversely, if it would be prevented by buttressing of ice shelves.

The relevant equation for vertical mean quantities at the grounding line, derived by simple force balance (Bassis and Walker, 2012; Pollard et al, 2015), is:

$$2\,\tau_{x'x'} = \frac{\rho_i(1-\rho_i/\rho_w)gh^2\theta}{2(h-d_s-d_b)} \tag{11}$$

where $\tau_{x'x'}$ is the depth-averaged normal deviatoric stress at the grounding line (in direction $x'$ to distinguish it from the model's $x$ axis). Note that this applies equally to grounding lines with ice shelves, and to ice cliffs at grounding lines without an ice shelf (for which $\theta$ =1). The crevasse depths $d_s$ (surface) and $d_b$ (basal) are Nye-depths as described in section 2.3 above, and depend on principal deviatoric stress. Their sum $d_s + d_s = \theta\,h\,/\,2$, where $h$ is the ice thickness (Pollard et al., 2015). $\theta$ in (11) is from Eq. 6b, using the principal stress direction yielding maximum $\theta$.

With $x'$ in the horizontal principal stress direction, the quantity $2\tau_{x'x'}$ is a good approximation for the difference in the two principal stresses in the $x'$ and $z$ plane, which is reported in laboratory experiments as a measure of ice yield strength, typically around ~1 MPa (Bassis and Walker, 2012). Several other considerations may modify this value and the concept of a uniform ice yield strength itself (Parizek et al., 2019; Clerc et al., 2019), including deformation unique to cliffs such as slumping and torques, ice cohesion and modes of failure depending on depth, and importantly, the amount of pre-existing fractures, buried crevasses, bubbly ice and/or cm-scale grain sizes, as opposed to relatively pristine ice with small (~mm-scale) grain sizes. Ice with extensive pre-existing damage is prevalent in most ice cores and presumably throughout Antarctica, and has yield strengths around ~1 MPa, much weaker than pristine ice; Parizek et al. (2019) and Clerc et al. (2019) agree that maximum heights of

subaerial ice cliffs (above sea level, with ~9 times that below sea level) are approximately 100 to 200 m for pre-damaged ice, and
340   ~500 m for pristine ice. In our diagnosis below, we assume that central West Antarctic ice is typically pre-damaged (or if it is not
already, it will likely become so as the rapidly retreating grounding line approaches from the north), and so assume an ice yield
strength of around ~1 MPa.

In Fig. 12, $\theta$ and the relevant deviatoric stress measure ($2\tau_{x'x'}$ from Eq. 11) are plotted at grounding lines for the simulation
without hydrofracturing or cliff failure (type 2), and with the new modifications in section 2. For modern, $2\tau_{x'x'}$ is far below 1
345   MPa at all grounding lines, as it should be as no significant structural failure is observed today. At +400 years into the run
(~2350 CE), when the retreating central West Antarctic grounding lines are beginning to encounter deep (>1 km) bathymetry, the
surviving ice shelves shown in Fig. 9 still provide some buttressing, and most buttressing factors are well below 1; (even though
these ice shelves are too short and thin to reach distant pinning points, the lateral curvature in their flow produces back stress).
Most $2\tau_{x'x'}$ values are somewhat below 1 MPa, indicating that extensive structural failure is unlikely.

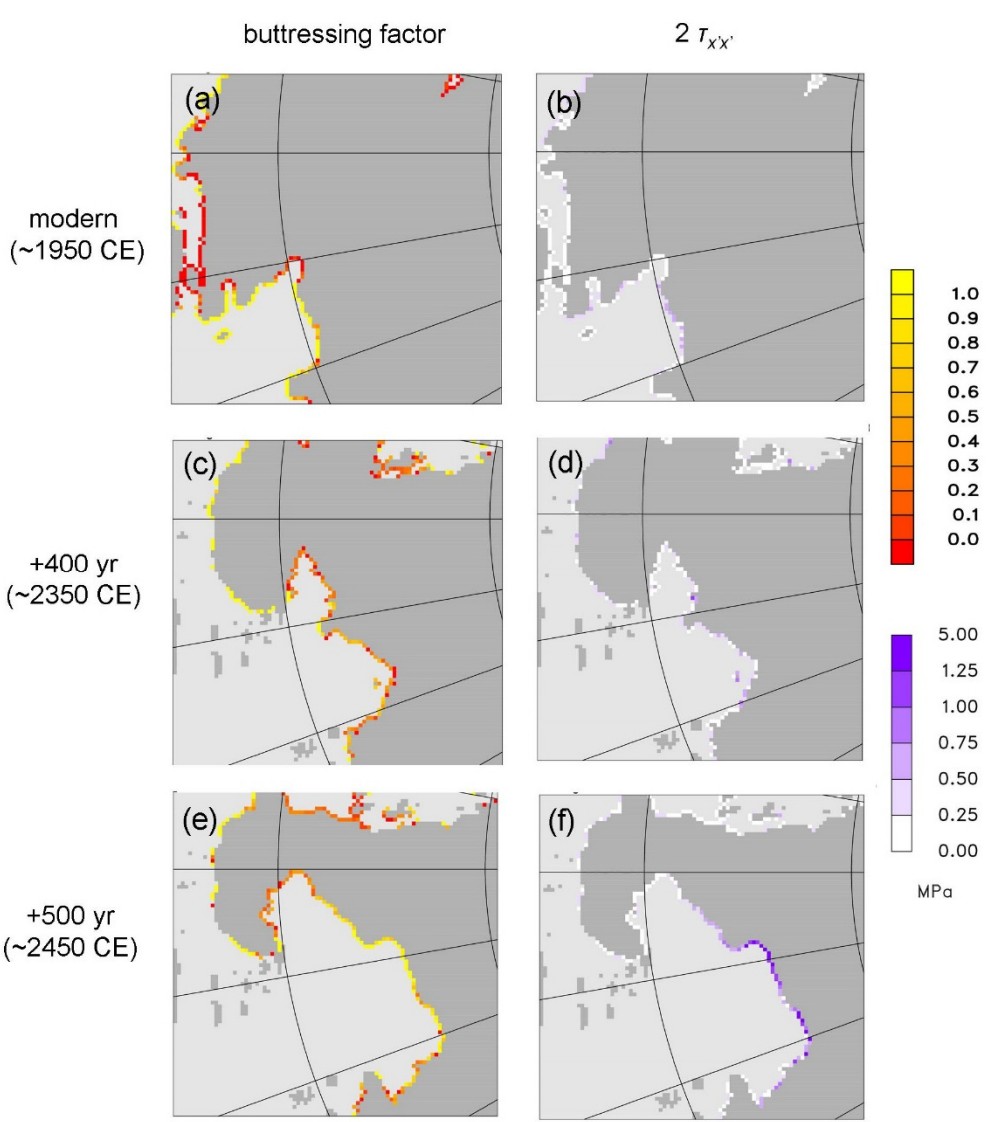

350

**Figure 12.** Grounding-line quantities in simulated future West Antarctic ice retreat with RCP8.5 forcing, without hydrofracturing or cliff failure, for the new model version. **1st row (a-b):** at year 0 (~1950 CE). **2nd row (c-d):** at year 400 (~2350 CE). **3rd row (e-f):** at year 500 (~2450 CE). **1st column (a,c,e):** buttressing factor $\theta$ at grounding line. **2nd column (b,d,f):** $2\tau_{x'x'}$, where $\tau_{x'x'}$ is the depth-averaged deviatoric normal stress at grounding lines, MPa (Eq. 11). An enlarged subset of the model domain is shown, to better show the grounding-line quantities in the central West Antarctic regions with deep bathymetry.

However, by +500 years (~2450 CE), central West Antarctic grounding lines experience even deeper bathymetry, and many $\theta$ values are at or close to 1 (weakly buttressed or essentially unbuttressed). Many $2\tau_{x'x'}$ magnitudes are at or exceed 1 MPa, indicating that structural failure of these grounding-line columns would occur.

## 7. Discussion and Conclusions

The modifications described above in calculating the 2-D orientation of the grounding line, imposed ice flow direction, and buttressing factor yield physically reasonable results. The first modification more realistically represents the true geometry of the grounding line.

In the idealized fjord-like MISMIP+ and MISMIP3d experiments, which involve strong 2-D curvature of grounding lines in a rectangular channel, the modifications have significant effects on the model's grounding-line variations, bringing them in line with those of other higher-order higher-resolution models in the intercomparisons (Cornford et al., 2020; Pattyn et al., 2013; Patty and Durand, 2013). Best overall intercomparison results are obtained with the buttressing factor at the grounding line based on the maximum extensional stress over all directions (Appendix A).

In contrast, the modifications have relatively little effect in large-scale simulations of future rapid West Antarctic ice retreat. This is presumably because of the larger lateral scales of major West Antarctic basins, so that grounding-line retreat in these basins is more one-dimensional in character, and better represented by the simpler "staircase" grounding-line treatment of the standard model. This is borne out by results of pan-Antarctic experiments in the ABUMIP intercomparison (Sun et al., 2020), which lie within the range of the other models.

The improved treatments of grounding-line orientation and buttressing factor allow us to better diagnose the force balance at grounding lines in the West Antarctic simulations, to see if structural failure could occur in a future with unmitigated greenhouse-gas warming. We find that when grounding lines reach very deep central West Antarctic regions (~1 to 2.5 km below sea level) after about 500 years, ice-shelf buttressing is weak and the deviatoric stress measures widely exceed the ice yield stress, implying that structural failure would occur at these grounding lines. In that case, a runaway disintegration could be initiated, with structural failure propagating very rapidly into the remaining grounded ice (Schlemm and Levermann, 2019), which in the absence of renewed buttressing would continue until shallower bathymetry is reached to the south.

Several other ice sheet-shelf models have performed similar projections of future West Antarctic retreat (e.g., Feldmann and Levermann, 2015; Golledge et al., 2015; Arthern and Williams, 2017), some with higher order and/or higher resolution than ours. We suggest it would be beneficial to examine these grounding-line quantities in other model simulations, to more robustly assess the danger of structural failure in future centuries under RCP8.5-like climate warming.

Apart from ice shelves, another potential source of buttressing is from mélange. Huge amounts of floating ice debris (mélange) would be generated in front of the retreating ice fronts in the above scenarios. In major Greenland fjords today such as Jakobshavn and Helheim, mélange is considered to provide significant back stress on the glacier calving front, at least in winter (e.g., Burton et al., 2018). However, in one study using a heuristic continuum model of mélange (Pollard et al., 2018), its back stress on ice shelves and grounding lines is negligible during West Antarctic retreat. In contrast to the narrow Greenland fjords, mélange in the much wider West Antarctic embayments flows northward into the Southern oceans nearly unimpeded.

Other processes that could reduce the deep bathymetry encountered by future grounding lines are bedrock rebound under the reduced ice load, and less gravitational attraction of the ocean by the receding ice (Gomez et al., 2015). The West Antarctic simulations here include the first process, using a relatively simple ELRA (Elastic Lithosphere Relaxing Asthenosphere) bed model (Pollard and DeConto, 2012), and the rebound of the modern bathymetry (Fig. 11) under the central grounding lines after 400 to 500 years (Fig. 12) is minor. However, recent geophysical data indicate very low mantle viscosities below parts of West Antarctica (Heeszel et al., 2016; Barletta et al., 2018), which could produce faster rebound and shallower bathymetry by the time grounding lines retreat into central regions. Work to develop Earth-sea level models with laterally varying properties and ice-ocean gravitational interaction, and couple them with ice-sheet models, is ongoing (Gomez et al., 2018; Powell et al., 2020).

## Appendix A. Variations in calculating $\theta$

For all new-model results in the main paper, the buttressing factor $\theta$ is given by Eq. (6b), in which the deviatoric normal stress at the grounding line is given by $N_{max}$, its maximum extensional (principal stress) value over all possible directions 0 to 360º in Eq. (4). It is a good approximation in the central part of fjord-like channels (Gudmundsson, 2013, Fig. 1); in the shearing margins with stronger buttressing, the resulting $\theta$ values still agree reasonably (ibid; his Fig. 2 vs. our Fig. 5). As shown below, this method yields better overall MISMIP+ and MISMIP3d results than all alternatives tried, including simply using the grounding-line normal $(n_x,n_y)$ in Eq. (4) and (6a). We suspect this is because during retreat of an otherwise uniform grounding line in our model, unavoidably there are isolated single grid-cell changes from grounded to floating to ice at each timestep. This produces temporary zig-zags in the grounding line that are not completely muted by the orientation algorithm, and cause spurious single-cell distortions of the flow and overall retreat if $\theta$ is given by (6a), which are avoided if $\theta$ is given by (6b). However, we emphasize that the latter method was chosen for the main paper not because of the above rationale, but because it yields the best overall intercomparison results.

Several alternate methods of determining $\theta$ are described below, and results are compared with the method using $N_{max}$ and Eq. (6b) as in the main paper. These alternatives stem from the inherent uncertainty in using a 1-D flowline parameterization (Eq. 1) within a 2-D model. Uncertainties in estimating $\theta$ in numerical models are also discussed in Gudmundsson (2013). The four alternate methods for calculating $N$ in Eq. (4) and hence $\theta$ in Eq. (6) are as follows, labelled A to D:

   A.  Using the direction $(n_x,n_y)$ normal to the grounding line given by the new orientation algorithm in section 2.
   B.  Using the direction of ice flow from the preliminary grid-solution $(u,v)$ (these velocities are also used in Eq. 5).
   C.  Using maximum $N$ over all directions (Eq. 6b) as in the main paper, but with the strain rates in Eq. (5) calculated for the first ice shelf cell that is entirely surrounded by other ice-shelf cells, searching along a trajectory $(n_x,n_y)$ normal to the

grounding line. This avoids "contaminating" the strain rates with velocity points within grounded ice, especially for ice-shelf cells with up to 3 neighboring grounded-ice cells.

D. Using maximum $N$ over all directions (Eq. 6b) as in the main paper, but with the parameterized speed $U_g$ in (1) applied in the direction normal to the grounding line $(n_x, n_y)$, and with the component parallel to the grounding line equal to that of the preliminary grid-solution (cf., Gudmundsson, 2013, Fig. 1).

MISMIP+ and MISMIP3d results for all four methods are shown in Fig. A1. For comparison, thin black lines in each panel show results for the method used in the main paper (Eq. 6b). For MISMIP+, methods A and B yield similar results to the main paper,
all within the shaded ranges of the other models. Method C diverges drastically for the Ice1 experiment (Fig. A1c), and method D is nearly outside the range for the Ice2 experiment (Fig. A1h).

For MISMIP3d, methods A and B yield poor results similar to our original MISMIP3d experiments, with considerably larger grounding-line excursions and quite different total changes than the other models. Methods C and D yield almost the same results as the main paper, much closer or within the other model ranges. Hence all four alternate methods A-D yield results that
are poorer than that in the main paper, for at least one of the MISMIP+ and MISMIP3d experiments.

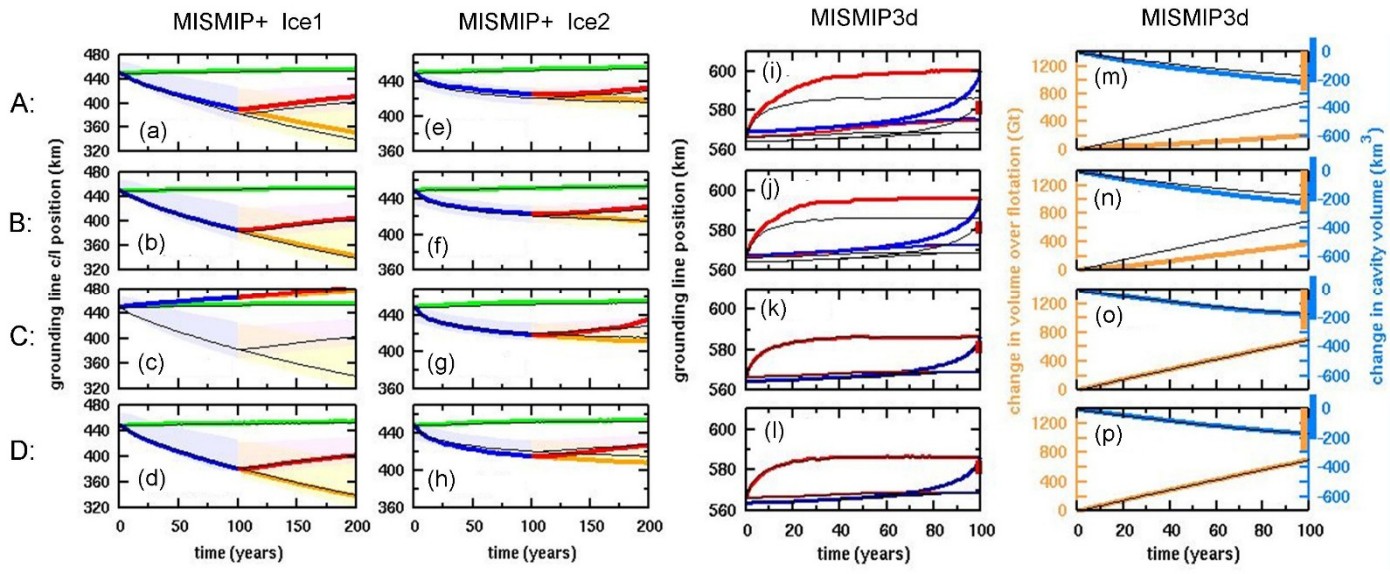

**Figure A1.** MISMIP+ and MISMIP3d results for four alternate methods of determining buttressing factor $\theta$. **Rows (top to bottom)** are for methods A to D described in the text. **Column (a-d):** MISMIP+ Ice1 experiment, as in Fig. 3. **Column (e-h):** MISMIP+ Ice2 experiment, as in Fig. 6. **Column (i-l):** MISMIP3d experiment, grounding-line positions as in Fig. 7a. **Column (m-p):** MISMIP3d experiment, changes in total
volume over flotation and cavity volume as in Fig. 7b. **Thick colored lines:** as in the main paper except with one of the alternate $\theta$ methods. **Thin black lines:** as in the main paper with the new $\theta$ method described in section 2 using Eq. (6b).

Fig. A2 shows simulations of future West Antarctic retreat, for all four alternate $\theta$ methods described above. The alternate methods have very little effect on equivalent sea-level rise, as also seen for the new vs. previous method in the main paper (Fig. 8); again this is presumably due to the more one-dimensional character of ice retreat in major Antarctic basins, with wider lateral
scales than the MISMIP+ and MISMIP3d channels.

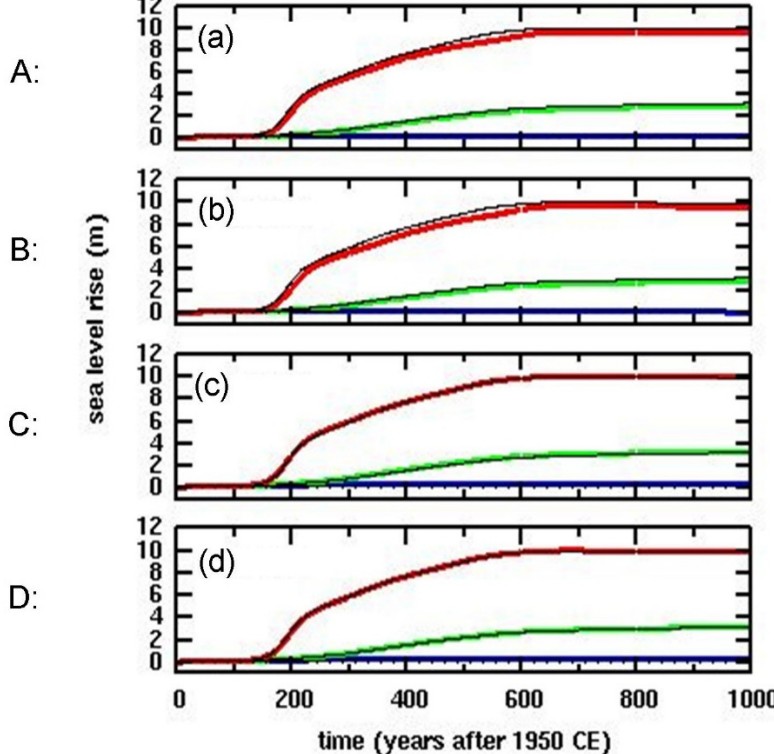

**Figure A2.** Equivalent global sea level rise in simulations of future West Antarctic ice retreat with climate forcing based on RCP8.5 greenhouse-gas scenario. **(a) to (d):** with the four alternate $\theta$ methods A to D described in the text. **Thick colored lines**: as in the main paper except with one of the alternate $\theta$ methods. **Thin black lines:** as in the main paper with the new $\theta$ method described in section 2. **Blue:** control (perpetual modern climate). **Green:** with RCP8.5 forcing, without hydrofracturing or cliff failure. **Red:** with RCP8.5 forcing, with hydrofracturing and cliff failure.

## Appendix B. Speculative modifications in grounding-line flux parameterization

Going beyond Schoof (2007), a few recent analytical studies have investigated aspects of boundary-layer treatments of grounding-line zones (Reese et al., 2018; Haseloff and Sergienko, 2018; Sergienko and Wingham, 2019). Here we briefly test three modifications to our grounding-line flux implementation that are more heuristic and speculative than those in the main paper. They are roughly motivated by the recent studies although they cannot represent them directly.

### B1. Strong buttressing

If the buttressing factor $\theta$ given by (6) falls to zero or below, this corresponds to compressive horizontal deviatoric stress normal to the grounding line, and compressive (negative) strain in the direction of flow. However, its use in the Schoof formation (1) for grounding-line ice velocity $U_g$ unrealistically predicts very small or zero $U_g$ as $\theta$ falls to zero. (Eq. (1a) would be invalid for $\theta <$ 0, as noted by Reese et al. (2018); for this equation we reset $\theta$ to be within the range [0,1] as mentioned in section 2). This does not occur extensively in our simulations of future Antarctic retreat, because buttressing is generally small as grounding lines rapidly recede into wide interior basins, and is more of a concern in colder climates with expanded grounding lines and shelf ice.

To crudely assess the problem, the value of $\theta$ in Eq. (1a) is adjusted for small values so that it does not fall exactly to 0.

$\qquad$ If $\theta < 0.3$, then $\theta' = \theta + 0.15\,(\,(0.3 - \theta)/0.3\,)^2$ $\qquad\qquad$ (B1)

(This is applied after $\theta$ is reset to the range [0,1] for Eq. (1a)). The adjusted value $\theta'$ falls only to 0.15 for strong buttressing, allowing small but non-zero flux. This does not rigorously address the problem, but can provide a guide to its severity by its effect on results.

## B2. Strain softening

This modification addresses the presumed underestimate of strain softening in the grounding zone in a purely 1-D flowline treatment such as Schoof (2007). With no lateral variations, the second invariant of the horizontal strain tensor, entering in ice viscosity in the SSA equations, is

$$\dot{\varepsilon}_{1D}{}^2 = (\partial u/\partial x)^2 \qquad\qquad (B2)$$

as in Schoof (2007). With lateral variations and two-dimensional flow, it is

$$\dot{\varepsilon}_{2D}{}^2 = (\partial u/\partial x)^2 + (\partial v/\partial y)^2 + (\partial u/\partial x)(\partial v/\partial y) + \frac{1}{4}(\partial u/\partial y + \partial v/\partial x)^2 \qquad (B3)$$

Then the ice viscosity $\eta$ is

$$\eta = \frac{1}{2\,A^{1/n}\,\dot{\varepsilon}^{(n-1)/n}} \qquad\qquad (B4)$$

(e.g., Thoma et al., 2014), where $\dot{\varepsilon}$ is either $\dot{\varepsilon}_{1D}$ or $\dot{\varepsilon}_{2D}$, and $A$ and $n$ are the rheological coefficient and exponent respectively appearing in the Schoof formula Eq. 1a. In our implementation Eq. (B2) and (B3) are computed using the velocity solution of the 475 previous iteration (Pollard and DeConto, 2012), at the last grounded cell adjacent to the grounding line.

$\eta$ does not enter in Eq. (1a), but we attempt to compensate for the absence of the 2-D strain-softening in (B4) by altering $A$ in Eq. (1a) by an appropriate factor:

$$A' = A\left(\frac{\dot{\varepsilon}_{2D}}{\dot{\varepsilon}_{1D}}\right)^{n-1} \qquad\qquad (B5)$$

This is not rigorous because the Schoof analysis incorporates the 1-D dependence (B2) in its derivation, and not (B3). However 480 the modification to $A$ in (B5) is at least in the right direction (increasing the ice flux across the grounding line), and may be useful as a crude approximation.

## B3. Overestimate of ice flux for high basal sliding coefficients

Sergienko and Wingham (2019) found that in ice streams with high basal sliding coefficients, the boundary-layer expansion of Schoof (2007) is not valid, and can overestimate the flux of ice across the grounding zone. Following on from that paper, the ratio of the newly calculated flux to the Schoof-calculated flux, in idealized tests for small basal slopes, ranges from ~0.6 to 1 but can be much smaller for steeper slopes (O. Sergienko, personal communication, 2020). This analysis cannot be represented by modifications in our model. However, we can crudely estimate the possible effect of such changes for small basal slopes at least, by simply reducing all imposed grounding-line velocities in (1) by a constant factor, i.e., multiplying $U_g$ given by (1b) by a factor 0.6.

## B4. Effects on results

The effects of applying each of the modifications described above are shown here. Fig. B1 shows results for the MISMIP+ Ice1 experiment, where the effects are similar in magnitude to those shown in the main paper (Figs. 3). By and large, the grounding-line excursions here are still within in the envelopes of other models in the MISMIP+ intercomparison (Cornford et al, 2020).

For the small-$\theta$ modification (section B1, Fig. B1a), the differences from the main-paper results are negligible, implying that the shortcomings of the flux parameterization (Eq. 1a) for strong buttressing do not have a large effect on grounding-line migration, at least in fjord-like scenarios. The reason may be that in regions of strong buttressing near the margins, grounding-line fluxes are relatively small, and allowing them to be zero has little effect on the overall evolution (consistent with Gudmundsson, 2013, Fig. 4). For the strain-softening modification (section B2, Fig. B1b), there is a serious degradation in results, which now are near the outer edges of the other model envelopes and exhibit spurious fluctuations, indicating this modification is not viable. For the 0.6 $U_g$ modification (section B3, Fig. B1c), the results are at least as good as in the main paper, implying that grounding-line migration is not extremely sensitive to uniform changes in the magnitude of the parameterized flux in Eq. (1a).

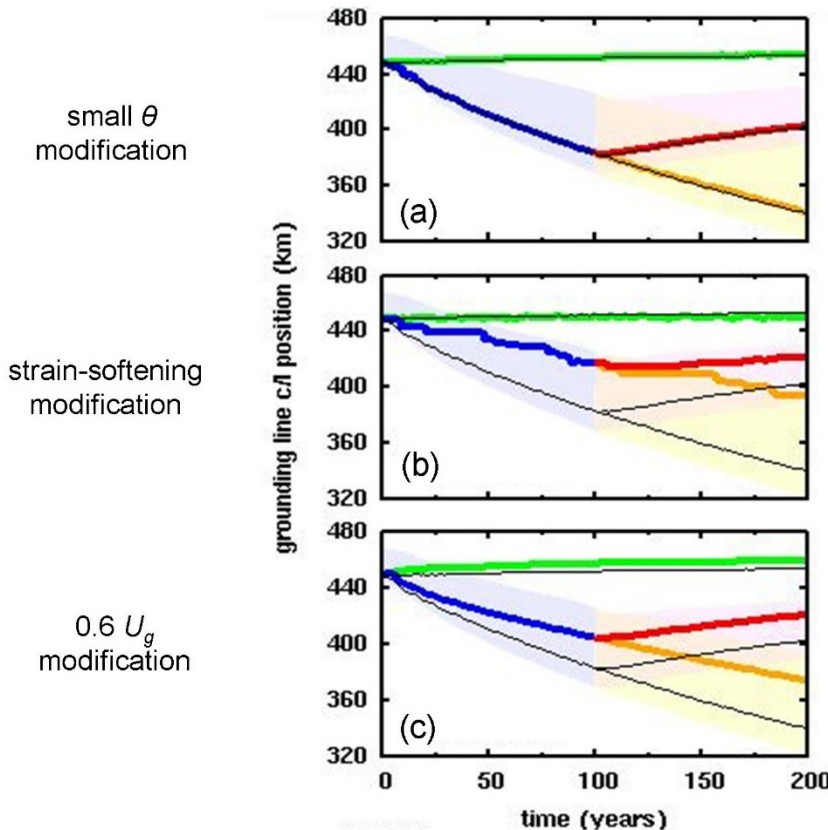

**Figure B1.** Along-fjord centerline position (km) of grounding lines in the MISMIP+ Ice1 experiment (Cornford et al., 2020). **(a)** with small-$\theta$ modification described in section B1. **(b)** with strain-softening modification described in section B2. **(c)** with 0.6 $U_g$ modification described in section B3**. Thick colored lines:** as in main paper except with one of the above modifications. **Thin black lines:** as in main paper with no further modification. **Green:** control (continuation of spin-up with zero oceanic melt). **Blue and yellow:** with oceanic melt perturbation. **Red:** with oceanic melt reset zero after year 100. Shaded regions show the envelopes for the "main subset" of MISMIP+ models, copied from Cornford et al. (2020, their Fig. 7a).

Fig. B2 shows results for future West Antarctic retreat, for simulations without hydrofracturing or cliff failure. All three modifications described above have very little effects on equivalent sea-level rise, as was also seen in the main paper (Fig. 8); again this is presumably due to the more one-dimensional character of ice retreat in major Antarctic basins, which have wider lateral scales than the MISMIP+ and MISMIP3d channels.

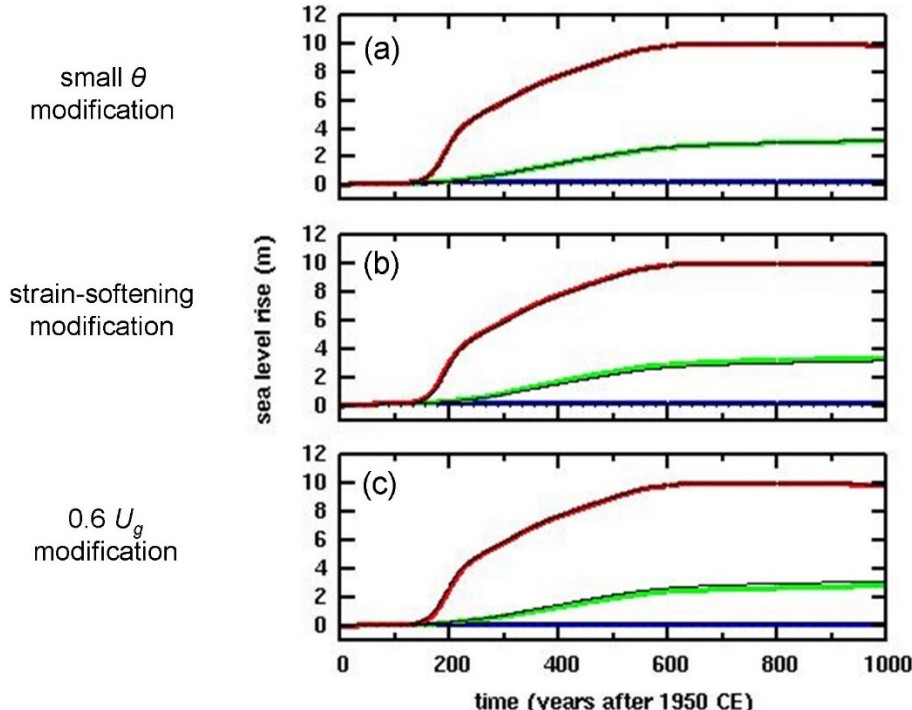

**Figure B2.** Equivalent global sea level rise in simulations of future West Antarctic ice retreat with climate forcing based on RCP8.5 greenhouse-gas scenario. **(a)** with small-$\theta$ modification described in section B1. **(b)** with strain-softening modification described in section B2. **(c)** with 0.6 $U_g$ modification described in section B3. **Thick colored lines:** as in main paper except with one of the above modifications. **Thin black lines:** as in main paper with no further modification. **Blue:** control (perpetual modern climate). **Green:** with RCP8.5 forcing, without hydrofracturing or cliff failure. **Red:** with RCP8.5 forcing, with hydrofracturing and cliff failure.

## Appendix C. Calculation of crevasse depths

For all runs in this paper, an improvement is made in the parameterization of crevasse depths, used both in "normal" calving and also in the cliff-failure physics (Pollard and et al., 2015). Crevasse depths are set to the Nye-depth (at which total horizontal stress is zero for surface crevasses, or is equal to water pressure for basal crevasses; Nye, 1957; Jezek, 1984; Nick et al., 2010). Previously, the divergence ($\partial u/\partial x + \partial v/\partial y$) was used along with ice viscosity as a simple estimate of the horizontal deviatoric stress (Pollard et al., 2015). Here, this is replaced by the maximum principal deviatoric stress (Turcotte and Schubert, 1982), calculated from the strain rates and viscosity. This is a small improvement "in principle". It has no effect in the idealized fjord MISMIP+ and MISMIP3d experiments for which calving is disabled, and has negligible effect in the West Antarctic simulations as shown in Fig. C1.

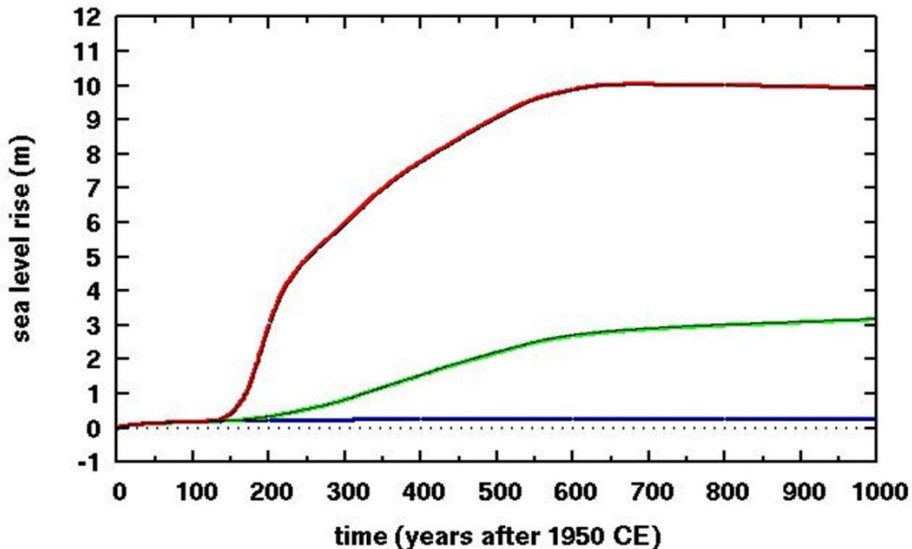

**Figure C1.** Equivalent global sea level rise in simulations of future West Antarctic ice retreat with climate forcing based on RCP8.5 greenhouse-gas scenario. **Thick colored lines:** as in main paper except with the previous parameterization of crevasse depths based on divergence. **Thin black lines:** as in the main paper (which includes the new crevasse-depth parameterization). **Blue:** control (perpetual modern climate). **Green:** with RCP8.5 forcing, without hydrofracturing or cliff failure. **Red:** with RCP8.5 forcing, with hydrofracturing and cliff failure.

*Code and data availability.* Selected output files, metadata and model code are available on Penn State's Data Commons archive at http://www.datacommons.psu.edu/commonswizard/MetadataDisplay.aspx?Dataset=6238, and at https://doi.org/10.26208/m3bt-jy63.

*Author contributions.* DP and RD conceived the project and design. DP performed coding and simulations and wrote the manuscript with input from RD.

*Competing interests.* The authors declare that they have no conflict of interest.

*Acknowledgements.* This work was supported by US National Science Foundation grant NSF ICER-1663693 and National Aeronautics and Space Administration grant NASA NNH16ZDA001N-SLCST. We thank Richard Alley for helpful advice on the discussion of ice yield stress in section 6, and reviewers Frank Pattyn and Stephen Cornford for insightful comments and suggestions.

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
