# Peer review of "Improvements in one-dimensional grounding-line parameterizations in an ice-sheet model with lateral variations (PSUICE3D v2.1)"

_Geoscientific Model Development, 2020_

## Short Comment (SC1) · 25 May 2020

Dear authors,

in my role as Executive editor of GMD, I would like to bring to your attention our Editorial version 1.2:

https://www.geosci-model-dev.net/12/2215/2019/

This highlights some requirements of papers published in GMD, which is also available on the GMD website in the 'Manuscript Types' section: http://www.geoscientific-model-development.net/submission/manuscript_types.html

[Figure]

In particular, please note that for your paper, the following requirement has not been met in the Discussions paper:

- "The main paper must give the model name and version number (or other unique identifier) in the title."

- Code must be published on a persistent public archive with a unique identifier for the exact model version described in the paper or uploaded to the supplement, unless this is impossible for reasons beyond the control of authors. All papers must include a section, at the end of the paper, entitled "Code availability". Here, either instructions for obtaining the code, or the reasons why the code is not available should be clearly stated. It is preferred for the code to be uploaded as a supplement or to be made available at a data repository with an associated DOI (digital object identifier) for the exact model version described in the paper. Alternatively, for established models, there may be an existing means of accessing the code through a particular system. In this case, there must exist a means of permanently accessing the precise model version described in the paper. In some cases, authors may prefer to put models on their own website, or to act as a point of contact for obtaining the code. Given the impermanence of websites and email addresses, this is not encouraged, and authors should consider improving the availability with a more permanent arrangement. Making code available through personal websites or via email contact to the authors is not sufficient. After the paper is accepted the model archive should be updated to include a link to the GMD paper.

Therefore, firstly, add the ice-sheet models name and version number in the title upon revision of your manuscript. Secondly, your code availability section disagrees in several ways with the GMD requirements:

- statements that the code is available from the author upon request are no longer

tolerated in GMD. Please provide a permanent code access or the respective license issues, which prohibit a free provision of the code. In the latter case, state exactly, which requirements need to be met to get access to the code. An author email as only access point for a code is much too volatile.

• The data and the code need to be made available for the discussion phase: promises for the future are not tolerated anymore. Therefore, provide asap (in a comment) and in the revised paper version, the exact link where your data is available now.

• the official title of this section is "Code and data availability", there is no need to create a new name.

Yours, Astrid Kerkweg

---

## Author Comment (AC1) · 29 May 2020

The model name and version number will be included in the revised title as:

"Improvements in one-dimensional grounding-line parameterizations in an ice-sheet model with lateral variations (PSUICE3D v2)".

The model code and selected output files are now archived and available online, and will be described in the revised text section:

*Code and data availability.* Selected output files, metadata and model code are available on Penn State's Data Commons archive at

http://www.datacommons.psu.edu/commonswizard/MetadataDisplay.aspx?Dataset=6238, and at https://doi.org/10.26208/m3bt-jy63.
* * *

---

## Referee Comment (RC1) · Frank Pattyn (Referee) · 24 Jun 2020

The paper presents improvements on a heuristic for grounding-line flux calculations in large-scale ice sheet models. The model initially participated in ice sheet intercomparisons focusing on ideal cases of grounding line behaviour and these published results are now used to improve the algorithm dealing with grounding line motion. The paper definitely valorises the benefit of model intercomparisons that often point to discrepancies or even model errors in some cases. The paper is well written, easy to understand and to follow. However, the paper is technical and therefore of interest for modellers dealing with such type of parameterisations. I would suggest to enlarge the scope a bit

in the introduction and explain in some more detail the reasons why such algorithms are necessary, what their advantages and disadvantages are. It would also make a wider readership interested in the problems currently encountered in marine ice sheet modelling. I would also suggest to provide a sketch of the proposed simple algorithm for grounding-line direction calculation. Reading through it (page 3 and 4), I took pencil and paper and made a quick drawing. It helped a lot in my understanding.

I have one major remark/question: both improvements (the grounding line orientation and the weighting scheme on grid velocities) improve the model performance so that it fits within the overall group of models. To what extent is this a clever way of fitting your model to the other models? A way to shed a light on this is to perform the MISMIP3d experiment and compare the result with the same adjustments to the other participating models. As shown in Pattyn and Durand (2013) the heuristic model shows large advance and retreat of the grounding line compared to conventional SSA models at high resolution.

The description of the calculation of crevasse depths falls somehow out of the scope of the paper. It is a model improvement but keeps the attention away from the main message and evaluation of the algorithm. furthermore, there is no experimental work presented regarding this modification. I would suggest to leave it out and use it appropriately in a subsequent manuscript that employs the improvement (typically an appendix).

Minor remarks:

Figure 4: please use a different color scheme for the buttressing factor. It is far from obvious to distinguish the colors of the end-members. Why not a scheme similar to the one used in the left panel?

---

## Referee Comment (RC2) · Stephen Cornford (Referee) · 6 Jul 2020

This paper described a modification to the 'PSU3D' ice sheet model, which has been used to carry out a wide variety of Antarctic simulations and has been used to produce some of the highest profile results in that field. The model is perhaps the best known of a number that determine an ice flow velocity across the grounding line from a ana-lytic expression derived by Schoof (2007) that applies to 1D flows without buttressing, adapted in some way to higher dimensional flows with buttressing. The signal charac-teristic of these models is that they perform far better than conventional models (that do not make use of the analytic expression) at low resolution ($\sim 10$ km). Conventional

models must be run at far finer resolutions ($1$ km) to produce plausible results. There are still discrepancies between these groups of models when the conventional models are run at fine resolution, and the modification in this paper addresses the difference that was evident in the MISMIP+ model comparison.

I think this is a good paper that describes its methods well and shows clearly the impact of the modification. I recommend publication, but ask the authors to consider two points (see general comments)

**1   General Comments**

The modifications described have an impact on the MISMIP+ results (a narrow channel with strongly curved grounding line) but little impact on the (probably more interesting) Antarctic experiments. One interpretation (and the interpretation given here) is that the unmodified model was already computing the relevant quantities well enough . That could be the case. But there is another source of information on this point: the ABUMIP comparison, which is set in Antarctica. This is in review, but the authors of this paper are co-authors of that paper, so are aware of its results . It seems that PSU3D is 'in the envelope' there, as well (at least from the figures I have seen), which seems to be further evidence in support of the author's position. Perhaps it is simply premature to cite a paper that has yet to be published and is not in 'open review', but it seems a shame to miss out on that extra evidence (I see that Frank Pattyn has also reviewed this paper and of course knows the ABUMIP results much better than I, so he may have more to say on that, but I have not looked for the sake of an independent review)

There is quite a lot of material on brittle failure / cliff collapse (section 5). I don't disagree that ice sheet modellers should be taking these things seriously, but it seems also a bit tangential to the topic of the paper. I don't think it detracts from the paper in any serious sense.
**2   Specific Comments**

Abstract, L10 "...presumably because dynamics in the wider major Antarctic basins are "adequately represented by the model's previous simpler one-dimensional formulation". see general comments - ABUMIP seems to support the case too.

L27 "Here we implement a more rigorous," Rigorous (as in mathematical rigour) does not seem like the right word, lacking a formal analysis of error. Complete?

L177 (MISMIP+ experiments).  What is the value of $A$?  In MISMIP+, PSU3d used a quite different value from other models to place its initial grounding line. Is that still the case?

---

## Author Comment (AC2) · 21 Jul 2020

We thank the two reviewers for their careful and insightful comments and suggestions, and plan to respond to them fully.

First, as requested the model name and version number (PSUICE3D v2) will be added to the title, and full archive information will be given in the Code and data availability section.

In response to the major remark of reviewer 1 (Frank Pattyn), we have added results for the MISMIP3d intercomparison. This is a great suggestion and provides an addi-

tional test of our new model modifications. We are able to achieve improved results both for MISMIP+ and MISMIP3d relative to other models. In doing this we found that best overall results are obtained with an additional physical modification, by determining the buttressing factor as the least-buttressed value over all orientations (0 to 360 degrees) at each point (but still with the Schoof velocity direction determined by the new grounding-line orientation algorithm). Results with some alternate modifications along these lines will be given in Appendix A.

As suggested by reviewer 2 (Stephen Cornford), we will mention the reasonable results of the model in the ABUMIP intercomparison, consistent with the findings here for larger-scale Antarctic applications.

All minor suggestions of both reviewers will be implemented:

- More scope on grounding-line flux parameterization will be added in the introduction.

- A sketch of the new grounding-line orientation scheme will be added in Fig. 1.

- The minor improvement in crevasse-depth calculation will be moved to Appendix C, with a figure showing that results are affected insignificantly.

- The green-yellow color scale used for buttressing factors was chosen to match that in Furst et al. (2016), but can definitely be changed. As requested we will change it to a new scale (yellow-red), distinct from others in the paper to distinguish this variable from others.

- The term "rigorous" will be replaced by "physically complete" and "realistic".

- The value of the rheological coefficient A in our MISMIP+ experiments will be specified.

---

## Author Response (AR1)

**Response to reviews, and marked-up manuscript, for "Improvements in one-dimensional grounding-line parameterizations in an ice-sheet model with lateral variations (PSUICE3D v2.1)", by D. Pollard and R.M. DeConto.**

We thank the reviewers for their careful and helpful reviews. We agree with all their comments and have acted on all their specific suggestions for changes. Each comment and our response is described below, with the reviewers' text in blue, and our response in black with new manuscript text indented. After that, other changes in the manuscript are described, including our response to the other comment received. Line numbers refer to the revised manuscript with no track changes shown. Finally, a copy of the revised manuscript is included (starting on pg. 10), showing all track changes since the original manuscript submission.

The largest changes are in response to referee 1's suggestion to also compare results for the MISMIP3d intercomparison. This is an excellent suggestion, providing an additional valuable test of the new model modifications. In performing these tests, this led us to a new modification in the grounding-line orientation treatment, as described in the revised Methods section, a new section showing MISMIP3d results, and a new Appendix comparing MISMIP+ and MISMIP3d results for alternate model versions. This is outlined in more detail below.

*Referee 1 (F. Pattyn):*

The paper presents improvements on a heuristic for grounding-line flux calculations in large-scale ice sheet models. The model initially participated in ice sheet intercomparisons focusing on ideal cases of grounding line behaviour and these published results are now used to improve the algorithm dealing with grounding line motion. The paper definitely valorises the benefit of model intercomparisons that often point to discrepancies or even model errors in some cases. The paper is well written, easy to understand and to follow. However, the paper is technical and therefore of interest for modellers dealing with such type of parameterisations. I would suggest to enlarge the scope a bit in the introduction and explain in some more detail the reasons why such algorithms are necessary, what their advantages and disadvantages are. It would also make a wider readership interested in the problems currently encountered in marine ice sheet modelling.

Lines 19-34: A new opening paragraph is added, providing background and motivation for the use of grounding-line parameterizations in ice-sheet models:

Accurate modeling of long-term Antarctic Ice Sheet variations requires simulation of ice dynamics in the zone between grounded ice and floating ice shelves, and grounding-line retreat and advance over century and millennial year time scales. Realistic simulation of grounding-line migration is challenging, requiring either higher-order or full-Stokes dynamics (e.g., Seddick et al., 2012), or at least a hybrid combination of horizontally stretching flow (Shallow Shelf Approximation, predominant in shelves and streams) and vertically shearing flow (Shallow Ice Approximation, predominant in inland flow) (e.g., Bueler and Brown, 2009). In any case, sensitivity tests have found that without additional measures, the grounding zone needs to be resolved at fine horizontal resolution on the order of ~100 m to avoid large numerical errors in grounding-line movement (Schoof, 2007; Goldberg et al., 2009; Gladstone et al., 2010, 2012; Pattyn et al, 2012; Cornford et al., 2016). Even with adaptive mesh refinement (Cornford et al., 2013, 2015), long-term $O(10^4$ to $10^6$ year) continental-scale simulations are currently computationally infeasible with this approach. Alternately, the ice flux across grounding lines can

be parameterized using an analytic boundary-layer treatment (Schoof, 2007), and embedded in an ice-sheet model (Pollard et al., 2012), making long-term large-scale simulations feasible. This approach performs reasonably well in some idealized model intercomparisons (Docquier et al., 2011; Pattyn et al., 2012; c.f., Gudmundsson, 2013), but less well in others with smaller-scale transient experiments (Pattyn et al., 2013; Pattyn and Durant, 2013; Drouet et al., 2013; Cornford et al., 2020). In this paper we describe new modifications to the parameterized grounding-line flux approach, and show that they significantly improve model performance in some intercomparisons.

I would also suggest to provide a sketch of the proposed simple algorithm for grounding-line direction calculation. Reading through it (page 3 and 4), I took pencil and paper and made a quick drawing. It helped a lot in my understanding.

Line 75: A third panel is added in Fig. 1c providing this sketch. We think it is what the reviewer has in mind, and agree it helps in explaining the algorithm.

I have one major remark/question: both improvements (the grounding line orientation and the weighting scheme on grid velocities) improve the model performance so that it fits within the overall group of models. To what extent is this a clever way of fitting your model to the other models? A way to shed a light on this is to perform the MISMIP3d experiment and compare the result with the same adjustments to the other participating models. As shown in Pattyn and Durand (2013) the heuristic model shows large advance and retreat of the grounding line compared to conventional SSA models at high resolution.

This is an excellent suggestion that had not occurred to us before, and provides an additional valuable test of the new model modifications. In performing these tests, they led us to a new modification in the grounding-line orientation treatment, which produces significantly improved results both for MISMIP+ and MISMIP3d. The new modification involves the calculation of buttressing at the grounding line using the maximum extensional (principal) deviatoric stress over all angles, while still using the grounding-line normal for the direction of the flux. It is described in section 2, lines 138-148:

In Appendix A, results are shown for several variations in calculating $N$ in (4) and $\theta$ in (6a). These alternatives stem from the inherent uncertainty in using a 1-D flowline parameterization (Eq. 1) within a 2-D model, and we use the MISMIP+ and MISMIP3d results as an empirical guide. The best overall intercomparison results are obtained not with the above method using the single direction $(n_x, n_y)$ in (4), but using the maximum extensional (principal) stress $N_{max}$, i.e., the maximum of $N$ over all possible directions 0 to 360°, and then

$$\theta = \frac{N_{max}}{\rho_i(1-\rho_i/\rho_w)gh/2} \qquad (6b)$$

For all new-model results in the main paper, $N_{max}$ is used and $\theta$ is given by (6b). A rationale for this method is discussed in Appendix A, but we emphasize that the choice is guided mainly because it yields the best overall MISMIP+ and MISMIP3d results among all variations tried (Fig. A1). Note also that $N_{max}$ is used only at the grounding line. In the ice-shelf interior, $\theta$ has no effect on the model physics, and where it is shown diagnostically below, the ice velocity at each point provides the orientation in (4).

Note that this new modification is used for all "new" results shown in the main paper, as mentioned in the text (line 72). The main results for MISMIP3d are described in new section 4, on lines 225-241:

The MISMIP3d intercomparison (Pattyn et al., 2013) offers another useful test of the new model versions. It uses a rectangular fjord-like setting as in MISMIP+, but with a uniformly sloping bed and perturbations in basal sliding coefficient instead of ocean melting. The models are first run to equilibrium, then the basal sliding coefficient is increased (slipperier bed) in a central region for 100 years causing the grounding line to advance, after which the perturbation is removed. Similarly to MISMIP+, our previous model produced larger and more rapid grounding-line advances than most other higher-order and/or higher-resolution models in the intercomparison (Pattyn et al., 2013), and consequently the changes in total volume over flotation and cavity volume differed from most models (Pattyn and Durand, 2013).

Fig. 7a,b shows the main results for the MISMIP3d experiment, for the new model version (solid lines) and the previous standard version close to that used in the original intercomparison. The centerline grounding line excursions in Fig. 7a for the new model version are considerably less than previously (~20 km vs. ~30 km), and much closer to the range of other model categories (red bar on the y-axis, from Pattyn and Durand, 2013). Notably, the equilibrated starting position of the grounding line is now around 560 km, much closer to those of most higher-order models in the intercomparison (~540 km, Pattyn et al., 2013; Pattyn and Durand, 2013).Changes in total volume over flotation and cavity volume in Fig. 7b are also much closer to the ranges of the other model categories (yellow and blue bars on the y-axis; Pattyn and Durand, 2013).

For completeness, spatial maps of changes in surface speed and elevation are shown in Figs. 7c-f, which can be compared with the same quantities for other model categories in Pattyn and Durand (2013) Figs. 2 and 3. There are some differences but the overall features and amplitudes are similar.

Fig. 7 shows our MISMIP3d results, using most of the same types of figures as in Pattyn et al. (2013) and Pattyn and Durand (2013).

In new Appendix A, the main new model version is compared with four alternate modifications in the buttressing calculation. This is new material, and as mentioned in the new text, the modifications stem from the inherent uncertainty in using a 1-D flowine parametrization in a 2-D model. The main point of this Appendix is to show that the "main" modification used in the main text yields the best overall MISMIP+ and MIPSMIP3d results, as shown in Fig. A1. Lines 398-429 (Appendix A) are:

For all new-model results in the main paper, the buttressing factor $\theta$ is given by Eq. (6b), in which the deviatoric normal stress at the grounding line is given by $N_{max}$, its maximum extensional (principal stress) value over all possible directions 0 to 360° in Eq. (4). It is a good approximation in the central part of fjord-like channels (Gudmundsson, 2013, Fig. 1); in the shearing margins with stronger buttressing, the resulting $\theta$ values still agree reasonably (ibid; his Fig. 2 vs. our Fig. 5). As shown below, this method yields better overall MISMIP+ and MISMIP3d results than all alternatives tried, including simply using the grounding-line normal ($n_x, n_y$) in Eq. (4) and (6a). We suspect this is because during retreat of an otherwise uniform grounding line in our model, unavoidably there are isolated single grid-cell changes from grounded to floating to ice at each timestep. This produces temporary zig-zags in the grounding line that are not completely muted by the orientation algorithm, and cause spurious single-cell distortions of the flow and overall retreat if $\theta$ is given by (6a), which are avoided if $\theta$ is given

by (6b). However, we emphasize that the latter method was chosen for the main paper not because of the above rationale, but because it yields the best overall intercomparison results.

Several alternate methods of determining $\theta$ are described below, and results are compared with the method using $N_{max}$ and Eq. (6b) as in the main paper. These alternatives stem from the inherent uncertainty in using a 1-D flowline parameterization (Eq. 1) within a 2-D model. Uncertainties in estimating $\theta$ in numerical models are also discussed in Gudmundsson (2013). The four alternate methods for calculating $N$ in Eq. (4) and hence $\theta$ in Eq. (6) are as follows, labelled A to D:

A. Using the direction $(n_x, n_y)$ normal to the grounding line given by the new orientation algorithm in section 2.

B. Using the direction of ice flow from the preliminary grid-solution $(u, v)$ (also used in Eq. 5).

C. Using maximum $N$ over all directions (Eq. 6b) as in the main paper, but with the strain rates in Eq. (5) calculated for the first ice shelf cell that is entirely surrounded by other ice-shelf cells, looking along a trajectory $(n_x, n_y)$ normal to the grounding line. This avoids "contaminating" the strain rates with velocity points within grounded ice, especially for ice-shelf cells with up to 3 neighboring grounded-ice cells.

D. Using maximum $N$ over all directions (Eq. 6b) as in the main paper, but with the parameterized speed $U_g$ in (1) applied in the direction normal to the grounding line $(n_x, n_y)$, and with the component parallel to the grounding line equal to that of the preliminary grid-solution (cf., Gudmundsson, 2013, Fig. 1).

MISMIP+ and MISMIP3d results for all four methods are shown in Fig. A1. For comparison, thin black lines in each panel show results for the method used in the main paper (Eq. 6b). For MISMIP+, methods A and B yield similar results to the main paper, all within the shaded ranges of the other models. Method C diverges drastically for the Ice1 experiment (Fig. A1c), and method D is nearly outside the range for the Ice2 experiment (Fig. A1h).

For MISMIP3d, methods A and B yield poor results similar to our original MISMIP3d experiments, with considerably larger grounding-line excursions and quite different total changes than the other models. Methods C and D yield almost the same results as the main paper, much closer or within the other model ranges. Hence all four alternate methods A-D yield results that are poorer than that in the main paper, for at least one of the MISMIP+ and MISMIP3d experiments.

The description of the calculation of crevasse depths falls somehow out of the scope of the paper. It is a model improvement but keeps the attention away from the main message and evaluation of the algorithm. furthermore, there is no experimental work presented regarding this modification. I would suggest to leave it out and use it appropriately
in a subsequent manuscript that employs the improvement (typically an appendix).

This minor improvement is now moved to Appendix C, where it is described briefly and a new Fig. C1 shows that it makes negligible difference in the West Antarctic results. Lines 519-526 (Appendix C) are:

For all runs in this paper, an improvement is made in the parameterization of crevasse depths, used both in "normal" calving and also in the cliff-failure physics (Pollard and et al., 2015). Crevasse depths are set to the Nye-depth (at which total horizontal stress is zero for surface crevasses, or is equal to water pressure for basal crevasses; Nye, 1957; Jezek, 1984; Nick et al., 2010). Previously, the divergence $(\partial u/\partial x + \partial v/\partial y)$ was used along with ice viscosity as a simple estimate of the horizontal deviatoric stress (Pollard et al., 2015). Here, this is replaced by the maximum principal deviatoric stress (Turcotte

and Schubert, 1982), calculated from the strain rates and viscosity. This is a small improvement "in principle". It has no effect in the idealized fjord MISMIP+ and MISMIP3d experiments for which calving is disabled, and has negligible effect in the West Antarctic simulations as shown in Fig. C1.

155

Minor remarks:

Figure 4: please use a different color scheme for the buttressing factor. It is far from obvious to distinguish the colors of the end-members. Why not a scheme similar to the one used in the left panel?

160      A new color scheme is used in all 2-D maps showing the buttressing factor $\theta$ (in Figs. 4, 5, 10, 12). It is red-to-yellow, and we have checked with the reviewer that this is a good scheme. It is different from other color schemes in the paper to distinguish the $\theta$ maps from those for other quantities.

*Referee 2 (S. Cornford):*

165

This paper described a modification to the 'PSU3D' ice sheet model, which has been used to carry out a wide variety of Antarctic simulations and has been used to produce some of the highest profile results in that field. The model is perhaps the best known of a number that determine an ice flow velocity across the grounding line from a analytic expression derived by Schoof (2007) that applies to 1D flows without buttressing, adapted in some way to higher dimensional flows with buttressing. The

170   signal characteristic of these models is that they perform far better than conventional models (that do not make use of the analytic expression) at low resolution (10 km). Conventional models must be run at far finer resolutions (1 km) to produce plausible results. There are still discrepancies between these groups of models when the conventional models are run at fine resolution, and the modification in this paper addresses the difference that was evident in the MISMIP+ model comparison.

175   I think this is a good paper that describes its methods well and shows clearly the impact of the modification. I recommend publication, but ask the authors to consider two points (see general comments)

**1 General Comments**

The modifications described have an impact on the MISMIP+ results (a narrow channel with strongly curved grounding line) but

180   little impact on the (probably more interesting) Antarctic experiments. One interpretation (and the interpretation given here) is that the unmodified model was already computing the relevant quantities well enough . That could be the case. But there is another source of information on this point: the ABUMIP comparison, which is set in Antarctica. This is in review, but the authors of this paper are co-authors of that paper, so are aware of its results . It seems that PSU3D is 'in the envelope' there, as well (at least from the figures I have seen), which seems to be further evidence in support of the author's position. Perhaps it is

185   simply premature to cite a paper that has yet to be published and is not in 'open review', but it seems a shame to miss out on that extra evidence (I see that Frank Pattyn has also reviewed this paper and of course knows the ABUMIP results much better than I, so he may have more to say on that, but I have not looked for the sake of an independent review).

As suggested, we note that the insensitivity of continental-scale Antarctic results to the modifications here is basically consistent

190   with the "within-envelope" behaviour of the model in ABUMIP. This is mentioned in section 5 on West Antarctic results, lines 288-290:

This is consistent with our results in the ABUMIP intercomparison involving continental Antarctic experiments, where the previous model version was used and results lie within the ranges of the other models (Sun et al, 2020).\

and in the concluding section on lines 370-371:

This is borne out by results of pan-Antarctic experiments in the ABUMIP intercomparison (Sun et al., 2020), which lie within the range of the other models.

There is quite a lot of material on brittle failure / cliff collapse (section 5). I don't disagree that ice sheet modellers should be taking these things seriously, but it seems also a bit tangential to the topic of the paper. I don't think it detracts from the paper in any serious sense.

This material is retained without modification (now in renumbered section 6). We think that it is suitable and worthwhile here because its validity is a direct consequence of the improved grounding-line treatment of the paper; i.e., our projections of the future potential for brittle failure in West Antarctica are considerably more robust due to the improved treatment. It demonstrates how idealized intercomparison tests can lead to improved model physics that are central to important real-world applications.

**2 Specific Comments**

Abstract, L10 "...presumably because dynamics in the wider major Antarctic basins are "adequately represented by the model's previous simpler one-dimensional formulation". see general comments - ABUMIP seems to support the case too.

As noted above, the ABUMIP connection is mentioned in two places in the new text. We do not mention it in the abstract to keep the abstract focused and brief.

L27 "Here we implement a more rigorous," Rigorous (as in mathematical rigour) does not seem like the right word, lacking a formal analysis of error. Complete?

We agree, and have replaced "rigorous" in these places with "realistic" (line 44), "physically complete" (line 46), and "improved" (line 372).

L177 (MISMIP+ experiments). What is the value of A? In MISMIP+, PSU3d used a quite different value from other models to place its initial grounding line. Is that still the case?

Values of the ice rheologic coefficient $A$ in our new MISMIP+ runs are now specified in the caption of Fig. 3. In that figure, three runs are shown for the MISMIP+ Ice1 experiment, and the different $A$ values are discussed in new text on lines 180-186:

Different values of rheologic coefficient $A$ are used as noted in the caption, in order for the equilibrated grounding line at the start of each experiment to have nearly the same $x$-axis location (~455 km). With the previous model version and original MISMIP+ $A$ value (crosses), the grounding-line variations are close to those in our original MISMIP+ runs, significantly

faster and larger than other higher-order, higher-resolution models as shown in Cornford et al. (2020). With the same model version and a slightly different value of $A$ (thin lines), the results are within the other-model envelopes (background shading), but close to their outer edges; this dependence on $A$ in our model was not noticed before.

235

The Fig. 3 caption (lines 190-194) is:

**Figure 3.** Along-fjord centerline position along the $x$-axis (km) of grounding lines in the MISMIP+ Ice1 experiments (Cornford et al., 2020). **Thick colored lines:** new model version and rheologic coefficient $A = 3$ x $10^{-17}$ $Pa^{-3}$ $a^{-1}$. **Crosses:** previous model version and $A =$ 2.5 x $10^{-17}$ $Pa^{-3}$ $a^{-1}$. **Thin black lines:** previous model version and $A = 3.5$ x $10^{-17}$ $Pa^{-3}$ $a^{-1}$. **Green:** control, with zero oceanic melt. **Blue**
240  **and yellow:** with oceanic melt perturbation. **Red:** with oceanic melt reset to zero after year 100. Shaded regions show the envelopes for the "main subset" of MISMIP+ models, copied from Cornford et al. (2020, their Fig. 7a).

*Response to interactive comment (A. Kerkweg, Executive editor):*

245  As requested, the model name and version number are now included in the title: "Improvements in one-dimensional grounding-line parameterizations in an ice-sheet model with lateral variations (PSUICE3D v2.1)". The model code and selected output files are archived and available online, as described in the renamed text section on lines 534-536:

*Code and data availability.* Selected output files, metadata and model code are available on Penn State's Data Commons
250  archive at http://www.datacommons.psu.edu/commonswizard/MetadataDisplay.aspx?Dataset=6238, and at https://doi.org/10.26208/m3bt-jy63.

Soon after our initial submission to GMDD, we uploaded the original model code and output files corresponding to that submission into that archive. We have now uploaded a new set of code and output corresponding to the revised paper, preserving
255  the original set in its own subdirectory as explained in the archive.

*Other changes*

The old modification involving "grid-cell weighting of imposed grounding-line velocities" described in section 2.2 of the
260  previous paper is removed, as it was not as well motivated or robust as the others, and would not contribute much to the new paper. (It is replaced by a somewhat analogous modification in the new Appendix B).

The three model versions used for MISMIP+ experiments in the main text of the previous paper (A=old, B=new, C=new plus grid-cell weighting) are replaced in the new paper (old, old with different $A$, new), which are shown in new Figs. 3 and 6.
265

The previous paper's Appendix A showing more speculative modifications is replaced by new Appendix B. The three modifications described in new sections B1 to B3 are: (B1) "Strong buttressing", with small $\theta$ values prevented from falling to zero, which is related to the previous paper's "grid-weighting" modification removed here as noted above; (B2) "Strain softening", same as old modification A1; (B3) "Overestimate of ice flux…", multiplying $U_g$ by 0.6, same as old modification A2.
270  The new text for section B1 is (lines 451-462):

**B1. Strong buttressing**

If the buttressing factor $\theta$ given by (6) falls to zero or below, this corresponds to compressive horizontal deviatoric stress normal to the grounding line, and compressive (negative) strain in the direction of flow. However, its use in the Schoof formation (1) for grounding-line ice velocity $U_g$ unrealistically predicts very small or zero $U_g$ as $\theta$ falls to zero. (Eq. (1a) would be invalid for $\theta < 0$, as noted by Reese et al. (2018); for this equation we reset $\theta$ to be within the range [0,1] as mentioned in section 2). This does not occur extensively in our simulations of future Antarctic retreat, because buttressing is generally small as grounding lines rapidly recede into wide interior basins, and is more of a concern in colder climates with expanded grounding lines and shelf ice.

To crudely assess the problem, the value of $\theta$ in Eq. (1a) is adjusted for small values so that it does not fall exactly to 0.

If $\theta < 0.3$, then $\theta' = \theta + 0.15\left(\left(0.3 - \theta\right)/0.3\right)^2$ $\qquad\qquad$ (B1)

(This is applied after $\theta$ is reset to the range [0,1] for Eq. (1a)). The adjusted value $\theta'$ falls only to 0.15 for strong buttressing, allowing small but non-zero flux. This does not rigorously address the problem, but can provide a guide to its severity by its effect on results.

Like before, these three modifications are compared in new section B4, "Effects on results" (lines 489-500 and 508-511):

[revised manuscript text omitted]

previous version       new version

[Figure]

**Figure 89.** Spatial maps of simulated future West Antarctic ice retreat with RCP8.5 forcing, without hydrofracturing or cliff failure, showing grounded ice surface elevations (m, rainbow scale) and floating ice thicknesses (m, pink scale). **1st row (a-eb):** at year 0 (~1950 CE). **2nd row (d fc-d):** at year 400 (~2350 CE). **3rd row (e-fg-i):** at year 500 (~2450 CE). **1st column (a,c,ed,g):**  previous model version . **2nd column (b,de,fh):**  new model version.  **3rd column (c,f,i):**

675

previous version      new version

modern
(~1950 CE)

+400 yr
(~2350 CE)

[revised manuscript text omitted]

---

## Author Response (AR2)

**Response to re-review, and marked-up manuscript, for revised version of "Improvements in one-dimensional grounding-line parameterizations in an ice-sheet model with lateral variations (PSUICE3D v2.1)", by D. Pollard and R.M. DeConto.**

We thank reviewer Frank Pattyn for his second review, with several small corrections and a suggestion for Fig. 1. We have implemented all these as listed below. Line numbers refer to those in the new manuscript without tracking. Following that, a copy of the re-revised manuscript is included showing the track changes for these corrections.

*Referee 1 (F. Pattyn), second review:*

Line 32: In the reference Pattyn and Durand, 2013, the misspelled name Durant is corrected to Durand.

Line 62: As suggested, "additive more speculative" is changed to "additive and more speculative".

Figure 1c and line 80: As suggested, model grid lines are added in Fig. 1c. They look somewhat uneven in my pdf, but are even in the JPG file provided in the zipped figure file. At the end of the caption, we added "Typical model grid cells are shown by dashed lines."

Line 363: "MISMPI3d" is corrected to "MISMIP3d".

*Other change:*

Line 159: We changed "affects the effects of" to "influences the effects of".

[revised manuscript text omitted]